EMBO
reports

# Hippo pathway controls biopterin metabolism to shield adjacent cells from ferroptosis in lung cancer

Hao Li [1,2], Yohei Kanamori[1,10], Akihiro Nita[3,10], Ayato Maeda[1,3], Tianli Zhang [4,5], Kenta Kikuchi [6], Hiroyuki Yamada[7,8], Touya Toyomoto[4], Mohamed Fathi Saleh[1,3], Mayumi Niimura[1], Hironori Hinokuma[1,8], Mayuko Shimoda[1], Koei Ikeda[8], Makoto Suzuki[8], Yoshihiro Komohara[7,9], Daisuke Kurotaki[6,9], Tomohiro Sawa[4] & Toshiro Moroishi [1,3,9 ✉]

## Abstract

Recent advances in single-cell technologies have uncovered significant cellular diversity in tumors, influencing cancer progression and treatment outcomes. The Hippo pathway controls cell proliferation through its downstream effectors: yes-associated protein (YAP) and transcriptional co-activator with PDZ-binding motif (TAZ). Our analysis of human lung adenocarcinoma and murine models revealed that cancer cells display heterogeneous YAP/TAZ activation levels within tumors. Murine lung cancer cells with high YAP/TAZ activity grow rapidly but are sensitive to ferroptosis, a cell death induced by lipid peroxidation. In contrast, cells with low YAP/TAZ activity grow slowly but resist ferroptosis. Moreover, they protect neighbouring cells from ferroptosis, creating a protective microenvironment that enhances the tumor's resistance to ferroptosis. Mechanistically, inhibiting YAP/TAZ upregulates GTP cyclohydrolase 1 (GCH1), an enzyme critical for the biosynthesis of tetrahydrobiopterin (BH4), which functions as a secretory antioxidant to prevent lipid peroxidation. Pharmacological inhibition of GCH1 sensitizes lung cancer cells to ferroptosis inducers, suggesting a potential therapeutic approach. Our data highlights the non-cell-autonomous roles of the Hippo pathway in creating a ferroptosis-resistant tumor microenvironment.

**Keywords** Hippo Pathway; Ferroptosis; Biopterin; Cell Communication; Lung Cancer
**Subject Categories** Autophagy & Cell Death; Cancer

## Introduction

Tissues comprise cells with identical genomes and distinct characteristics. The multiplicity of cell types within tissues is fundamental for tissue organization and specialization. Recent advancements in single-cell resolution technologies have unveiled that individual cells within the same cell type display heterogeneous phenotypes in tissue (Stegle et al, 2015). Therefore, considerable attention has been given to comprehensively delineating cellular heterogeneity and its significance in tissue homeostasis and dysfunction. Accordingly, accumulating evidence suggests that intratumoral cancer cell heterogeneity is closely associated with tumor development and therapeutic outcomes (Michor and Polyak, 2010; Zhu et al, 2023). The phenotypic heterogeneity of cancer cells within tumors integrates inputs from the local microenvironment and genetic and epigenetic alterations (Easwaran et al, 2014; Halbrook et al, 2022; Li et al, 2016). As each cell senses its local microenvironment and responds accordingly, environmental cues may play a pivotal role in programming the heterogeneous properties of cells. Spatial and temporal perturbations in the oxygen and nutrient supply, cytokine and growth factor levels, and pH within tumors shape the diversity of cancer cell populations. Therefore, the heterogeneous behavior of cells reflects the microenvironmental variability. Elucidating the molecular mechanisms underlying the cellular responses to these contextual signals may provide insights for improving the clinical management of patients with cancer.

The Hippo intracellular signaling pathway integrates and transmits various upstream stimuli from the extracellular environment, including cell–cell contact, mechanical signals, energy stress, oxidative stress, and hypoxia (Misra and Irvine, 2018; Rausch and Hansen, 2020). These upstream stimuli activate the core large tumor suppressor 1 (LATS1) and LATS2 kinases, which phosphorylate and promote the cytoplasmic retention of the transcriptional

[1]Department of Molecular and Medical Pharmacology, Faculty of Life Sciences, Kumamoto University, 1-1-1 Honjo, Kumamoto 860-8556, Japan. [2]Department of Biomedical Sciences, School of Biological and Environmental Sciences, Kwansei Gakuin University, 1 Gakuen Uegahara, Sanda 669-1330, Japan. [3]Division of Cellular Dynamics, Medical Research Laboratory, Institute of Integrated Research, Institute of Science Tokyo, 1-5-45 Yushima, Tokyo 113-8510, Japan. [4]Department of Microbiology, Faculty of Life Sciences, Kumamoto University, 1-1-1 Honjo, Kumamoto 860-8556, Japan. [5]Center for Integrated Control, Epidemiology and Molecular Pathophysiology of Infectious Diseases, Akita University, 1-1-1 Hondo, Akita 010-8543, Japan. [6]Laboratory of Chromatin Organization in Immune Cell Development, International Research Center for Medical Sciences, Kumamoto University, 2-2-1 Honjo, Kumamoto 860-0811, Japan. [7]Department of Cell Pathology, Graduate School of Medical Sciences, Kumamoto University, 1-1-1 Honjo, Kumamoto 860-8556, Japan. [8]Department of Thoracic Surgery, Graduate School of Medical Sciences, Kumamoto University, 1-1-1 Honjo, Kumamoto 860-8556, Japan. [9]Center for Metabolic Regulation of Healthy Aging, Graduate School of Medical Sciences, Kumamoto University, 1-1-1 Honjo, Kumamoto 860-8556, Japan. [10]These authors contributed equally: Yohei Kanamori, Akihiro Nita. ✉E-mail: moroishi.toshiro@tmd.ac.jp

co-activators yes-associated protein (YAP) and transcriptional co-activator with a PDZ-binding motif (TAZ, also known as WWTR1). Inactivating LATS1/2 causes dephosphorylation and nuclear translocation of YAP/TAZ, thereby inducing target gene expression primarily by binding to the TEA domain family members of transcription factors (TEAD) (Meng et al, 2016; Zheng and Pan, 2019). Therefore, YAP/TAZ activity represents the major functional output of the Hippo pathway. Given that the Hippo pathway serves as a sensor for environmental signals, it is conceivable that tumor tissues comprise individual cancer cells that exhibit heterogeneous Hippo pathway activity. However, the biological contexts in which cancer cells display heterogeneous status of Hippo pathway and its significance remain to be elucidated.

Accumulating evidence suggests that the Hippo pathway plays a pivotal role in cancer biology (Piccolo et al, 2023), including lung cancer (Liang et al, 2024). Previous studies have convincingly established that the Hippo pathway regulates multiple processes during lung cancer progression, including malignant cell transformation, cancer cell growth, drug resistance, immune evasion, and metastasis (Baroja et al, 2024; Franklin et al, 2023; Moroishi et al, 2015a). However, these studies have primarily focused on the cell-autonomous roles of the Hippo pathway in cancer cells. Consequently, the non-cell-autonomous functions of the Hippo pathway, particularly in the context of reciprocal interactions between cancer cells with different Hippo pathway status, remain unknown. In the present study, we investigate the impact of Hippo pathway heterogeneity within tumor tissues on the growth and cell death resistance of cancer cells in the context of lung cancer. We found that intratumoral heterogeneity of YAP/TAZ activity correlates with worse prognosis in patients with lung adenocarcinoma. Murine lung cancer cells with low YAP/TAZ activity show slow growth but resistance to ferroptosis, a form of cell death associated with iron metabolism and characterized by lipid peroxidation (Jiang et al, 2021; Lei et al, 2024). In contrast, cancer cells with high YAP/TAZ activity are more potent in cell proliferation but sensitive to ferroptosis. Intriguingly, cancer cells with high YAP/TAZ activity confer resistance to ferroptosis when co-cultured with cancer cells with low YAP/TAZ activity cells, suggesting that ferroptosis resistance propagates within adjacent cancer cells. Our results suggest that intratumoral heterogeneity of the Hippo pathway establishes a symbiotic relationship to overcome lipid peroxidative stress in the tumor microenvironment.

# Results

## Patients with lung adenocarcinoma exhibiting YAP/TAZ heterogeneity had a worse prognosis

To investigate the impact of Hippo pathway status on cancer progression, we first evaluated the activity of the Hippo pathway in lung cancer cells within the tumor tissues of patients with lung adenocarcinoma. YAP/TAZ function as the main downstream effectors of the Hippo pathway. The activation status of YAP/TAZ can be evaluated by their protein abundance as well as cytoplasmic/nuclear localization, where nuclear localization generally associates with their functional output. Immunohistochemical analysis using

antibodies that recognize both YAP and TAZ revealed that YAP/TAZ expression was sparse in normal lung tissues, while diffuse positive signals were observed in the adenocarcinoma tissues (Fig. 1A). Additionally, YAP/TAZ expression showed heterogeneity within the same lung adenocarcinoma tissues. A homogenous YAP/TAZ expression pattern was detected in 149 out of 195 patients (76%). In contrast, a heterogeneous expression pattern, indicating the coexistence of cells with high YAP/TAZ activity (YAP/TAZ$^{high}$)—which demonstrated marked accumulation and nuclear localization of YAP/TAZ—and cells with low YAP/TAZ activity (YAP/TAZ$^{low}$)—which demonstrated reduced or cytoplasmic localization of YAP/TAZ—were observed in the cancer cells in the tumor tissues of 46 patients (24%) (Fig. 1A). Consecutive immunostaining for YAP/TAZ and a cancer cell marker cytokeratin (CK) AE1/AE3 revealed that CK$^+$ cancer cells exhibited both homogenous and heterogeneous expression patterns of YAP/TAZ within tumor tissues (Fig. 1B). The clinicopathological factors associated with homogeneous YAP/TAZ expression patterns included female sex, low Brinkman Index, low pathological stage, and low nuclear grade (Table EV1). However, YAP/TAZ expression was not associated with age or epidermal growth factor receptor (EGFR) mutation status. Notably, patients with heterogeneous YAP/TAZ expression showed shorter recurrence-free and cancer-specific survival times than those with homogenous YAP/TAZ expression (Fig. 1C). Collectively, these results suggest that heterogeneous YAP/TAZ activity in cancer cells contributes to the progression of lung adenocarcinoma.

## Cancer cells with low YAP/TAZ levels protect their surrounding cells from ferroptosis

Next, we investigated the expression pattern of YAP/TAZ in a mouse model of lung cancer. We transplanted Lewis lung carcinoma (LLC) cells into immunocompetent syngeneic mice (Fig. 2A) and found that YAP/TAZ expression also exhibited heterogeneity within the murine lung cancer tissues (Fig. 2B). To determine how heterogeneous YAP/TAZ activity impacts the tumor phenotype, a YAP/TAZ double knockout (dKO) cell line was established using the clustered regularly interspaced short palindromic repeat (CRISPR)/Cas9 system (Ran et al, 2013), and the phenotypic differences between YAP/TAZ$^{high}$ and YAP/TAZ$^{low}$ cells were investigated. Because YAP and TAZ are close homologs that mostly have overlapping functions in the Hippo pathway, both components were simultaneously deleted to investigate their loss of function in LLC cells. YAP/TAZ deletion was confirmed using DNA sequencing (Fig. EV1A) and immunoblotting (Fig. EV1B). A marked reduction in YAP/TAZ transcriptional activity was observed in YAP/TAZ dKO cells, as evidenced by decreased *Ankrd1* expression, a transcriptional target gene of YAP/TAZ (Fig. EV1C). Tumor growth is predominantly determined by the capacity of cancer cells to proliferate while suppressing cell death. Consistent with the well-established concept that YAP/TAZ activity is associated with cell proliferation (Li et al, 2022; Misra and Irvine, 2018), we observed reduced cell proliferation in YAP/TAZ dKO cells compared with WT cells. We then investigated the effect of YAP/TAZ inactivation on cancer cell death by treating LLC cells with Actinomycin D (inducing apoptosis (Liu et al, 2016)) (Fig. EV2A), a combination of tumor necrosis factor-alpha (TNF-α), cycloheximide and Z-VAD-FMK (TCZ; inducing

**A**

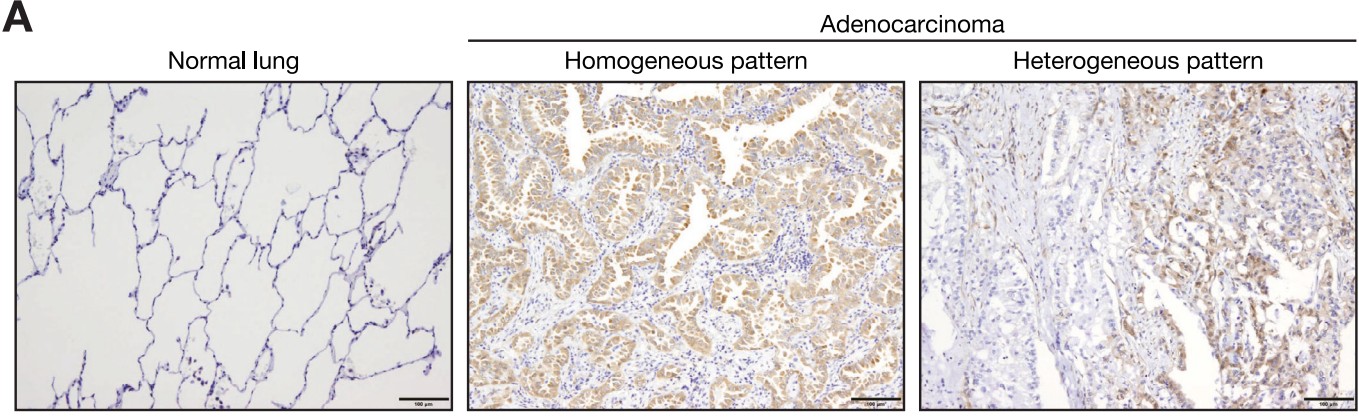

Normal lung

Adenocarcinoma

Homogeneous pattern

Heterogeneous pattern

**B**

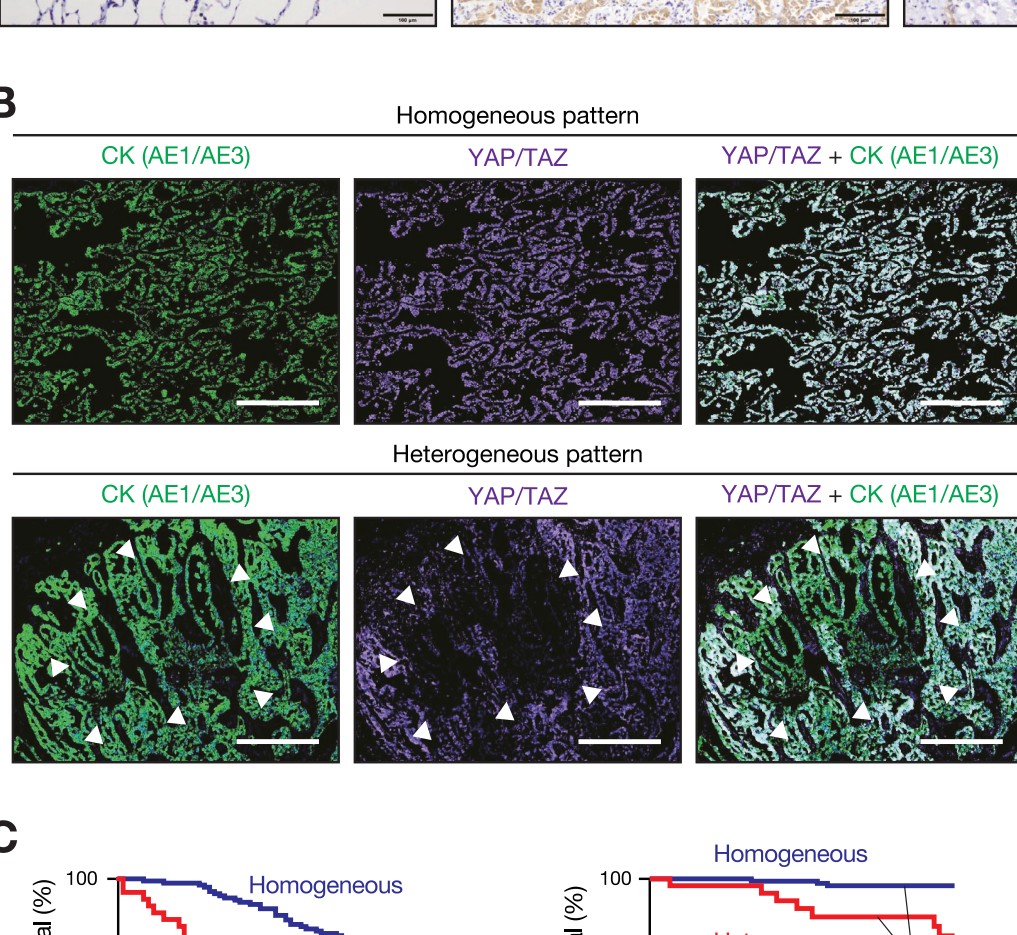

Homogeneous pattern

CK (AE1/AE3)   YAP/TAZ   YAP/TAZ + CK (AE1/AE3)

Heterogeneous pattern

CK (AE1/AE3)   YAP/TAZ   YAP/TAZ + CK (AE1/AE3)

**C**

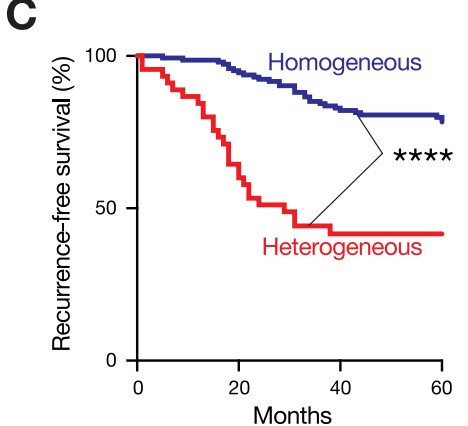

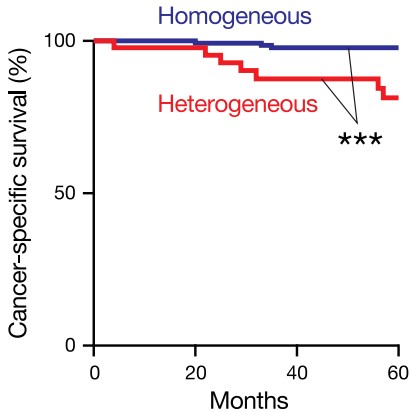

**Figure 1.  YAP/TAZ expression exhibits heterogeneity in lung cancer tissues, which is associated with a poorer prognosis.**

(A) Expression of yes-associated protein (YAP) and transcriptional co-activator with PDZ-binding motif (TAZ) (YAP/TAZ) exhibited heterogeneity in lung adenocarcinoma tissue. Immunohistochemical staining of YAP/TAZ was performed on paraffin-embedded tissues from patients with lung adenocarcinoma ($n = 195$), and representative images of homogenous and heterogeneous expression patterns were displayed. Scale bars, 100 μm. (B) Cancer cells exhibited both homogenous and heterogeneous expression patterns of YAP/TAZ within the same tumor tissues. Consecutive immunohistochemical staining of cytokeratin (CK) AE1/AE3 and YAP/TAZ was performed on paraffin-embedded tissues from patients with lung adenocarcinoma, with representative images shown. Green, CK. Purple, YAP/TAZ. Gray, overlap of CK and YAP/TAZ. The region enclosed by arrowheads highlights downregulation of YAP/TAZ. Scale bars, 500 μm. (C) Patients with YAP/TAZ heterogeneity showed worse recurrence-free and cancer-specific survival. Kaplan–Meier survival curves of patients with lung adenocarcinoma with or without YAP/TAZ heterogeneity (46 patients with heterogeneous YAP/TAZ expression patterns and 149 patients with homogenous YAP/TAZ expression patterns). ***$P = 0.000149575673274$; ****$P = 0.000000002704763$ (log-rank test). Source data are available online for this figure.

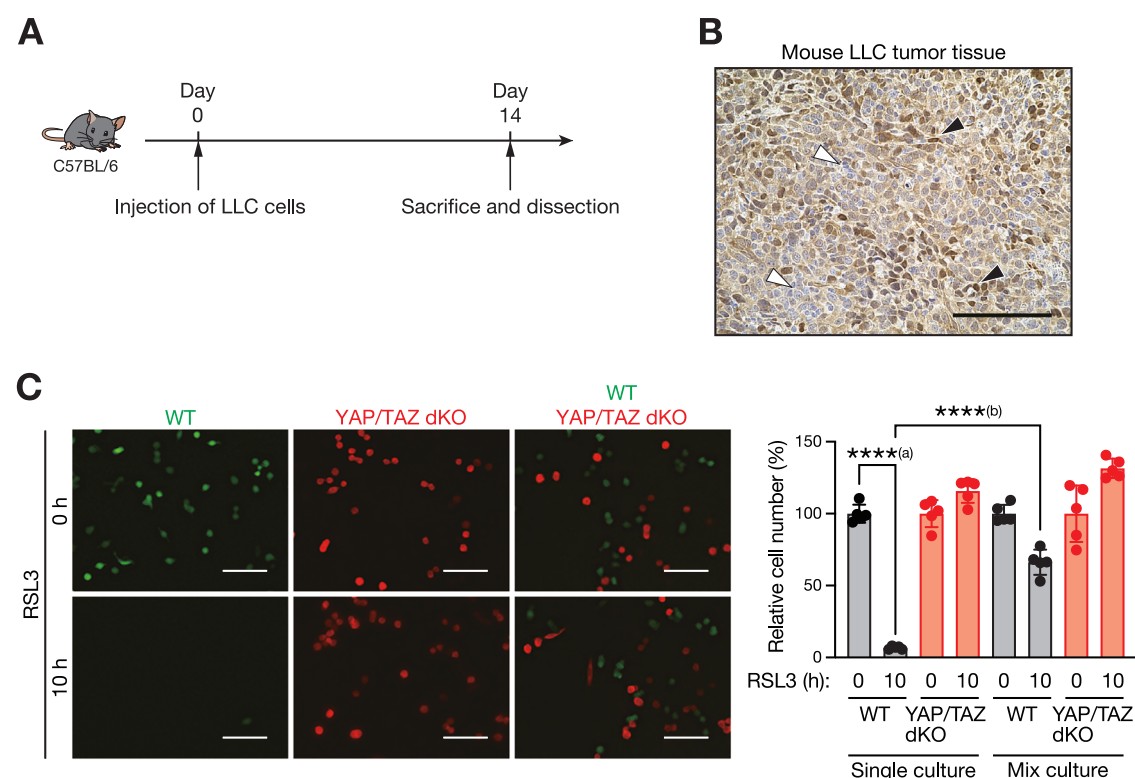

**Figure 2.  Cancer cells expressing low YAP/TAZ levels protect the surrounding cells from ferroptosis.**

(A) Schematic diagram of experimental design. Wild-type LLC lung cancer cells were subcutaneously injected into C57BL/6 mice, and the tumors were dissected after 14 days. (B) YAP/TAZ expression showed heterogeneity in mouse lung tumor tissues. Lewis lung carcinoma (LLC) cells were subcutaneously injected into C57BL/6 mice, and the resulting tumors were harvested 14 days after transplantation. Paraffin-embedded tumor tissues were subjected to immunohistochemical analysis using YAP/TAZ antibodies. Representative image from four tumors are shown. Black arrowheads indicate YAP/TAZ$^{high}$ cancer cells and white arrowheads indicate YAP/TAZ$^{low}$ cells. Scale bar, 100 μm. (C) LLC cells lacking YAP/TAZ protect their surrounding cells from ferroptosis. Wild-type (WT) LLC cells expressing EGFP (green) were co-cultured with YAP/TAZ double-knockout (dKO) LLC cells expressing tdTomato (red) in the presence of RSL3 (400 nM) for 10 h. Quantitative analysis of cell numbers is shown on the right. Data are presented as the means ± SD of five randomly selected images. The experiment was repeated three times, and data from one representative experiment are shown. Scale bars, 100 μm. ****$^{(a)}P = 0.000000000000105$; ****$^{(b)}P = 0.000000001145261$ (one-way ANOVA test followed by Tukey's multiple comparison test). Source data are available online for this figure.

necroptosis (Liu et al, 2018)) (Fig. EV2B), or RSL3 (inducing ferroptosis (Dixon et al, 2012)) (Fig. EV2C). We confirmed that cell death induced by these reagents was significantly suppressed by Z-VAD-FMK (an apoptosis inhibitor), Necrostatin-1 (a necroptosis inhibitor), or Ferrostatin-1 (a ferroptosis inhibitor), verifying the specificity of cell death induced by these chemicals. We found that while YAP/TAZ deletion sensitized LLC cancer cells to apoptosis (Fig. EV2D) and necroptosis (Fig. EV2E), YAP/TAZ loss

markedly conferred resistance to ferroptosis (Fig. EV2F). YAP/TAZ-deficient cells remained ferroptosis-resistant despite stimulation with high concentrations of RSL3 (Fig. EV2G). This is in line with the previous finding that YAP/TAZ confer resistance to apoptosis (Lai et al, 2011) while sensitizes the cells to ferroptosis (Wu et al, 2019; Yang et al, 2019; Yang et al, 2020). To validate our findings with different ferroptosis-inducing stimuli, cells were treated with several ferroptosis inducers, including Erastin, L-Buthionine-(S,R)-

Sulfoximine (L-BSO) and Sulfasalazine (SAS), or were cultured under cysteine and cystine-starved condition (CCS). We confirmed that cell death induced by these reagents was suppressed by the ferroptosis inhibitor Ferrostatin-1 (Fig. EV2H). YAP/TAZ depletion substantially suppressed cell death induced by these ferroptosis-inducing treatments (Fig. EV2I). Together, these data suggest that although YAP/TAZ inhibition suppresses lung cancer cell proliferation, it confers resistance to ferroptosis, which may contribute to the progression of tumors exhibiting heterogeneous activation statuses of YAP/TAZ.

To further characterize ferroptosis resistance observed in YAP/TAZ dKO cells, the lipid peroxidation levels, a key executive molecule triggering ferroptosis (Jiang et al, 2021), were investigated. While RSL3 treatment markedly induced lipid peroxidation in WT LLC cells, YAP/TAZ depletion strongly suppressed this induction (Fig. EV3A), suggesting that the Hippo pathway regulates ferroptosis upstream of lipid peroxide production. We confirmed the specificity of YAP/TAZ deletion by the CRISPR/Cas9 system, as re-expression of YAP (Fig. EV3B) restored transcriptional activity (Fig. EV3C) as well as RSL3-induced cell death in YAP/TAZ dKO cells (Fig. EV3D). Additionally, similar suppression of ferroptosis upon YAP/TAZ depletion was observed in another lung cancer cell line. Among the human lung cancer cell lines tested—A549, LK-2, PC-9, and H1975—LK-2 cells exhibited relatively higher YAP activity (Fig. EV3E), as determined by Phos-tag immunoblotting, which separates YAP proteins based on their phosphorylation status, with more heavily phosphorylated forms migrating more slowly. Consequently, LK-2 cells were selected for YAP/TAZ deletion experiments. Consistent with our findings in LLC cells, loss of YAP/TAZ reduced ferroptosis sensitivity in LK-2 cells (Fig. EV3F,G). These results convincingly confirmed the previous findings that the Hippo pathway alters cellular ferroptosis sensitivity (Wu et al, 2019; Yang et al, 2019; Yang et al, 2020). Although those studies have revealed the cell-autonomous functions of the Hippo pathway in regulating ferroptosis within cells, its non-cell-autonomous functions in affecting ferroptosis sensitivity in neighboring cells remain unknown. Given that our data indicate that YAP/TAZ heterogeneity within tumor tissues is associated with poor prognosis in patients with lung adenocarcinoma, we hypothesized that acquired ferroptosis resistance in YAP/TAZ^low cells may affect the properties of neighboring cells, contributing to tumor progression. To test this hypothesis, we conducted co-culture experiments with WT and YAP/TAZ dKO LLC cells. Strikingly, RSL3-induced cell death was suppressed in WT cells co-cultured with YAP/TAZ dKO cells compared to those cultured alone, suggesting that the acquired ferroptosis resistance in YAP/TAZ dKO cells propagated to the surrounding WT cells (Fig. 2C). Thus, collectively, our data indicate that YAP/TAZ inhibition within cancer cells suppresses ferroptosis induction and protects the surrounding cells from ferroptosis, thereby conferring multicellular-level protection against ferroptosis.

## Loss of YAP/TAZ induces GCH1 expression, the key enzyme for tetrahydrobiopterin biosynthesis

Although previous studies have consistently connected the Hippo pathway to cellular ferroptosis resistance, the proposed underlying mechanisms by which the Hippo pathway alters cellular ferroptosis sensitivity vary depending on the context and cell type (Wu et al, 2019; Yang et al, 2019; Yang et al, 2020). Indeed, validation experiments focusing on previously characterized molecules, such as TFRC, EMP1, ACSL4 and ANGPTL4, failed to elucidate the ferroptosis resistance observed in the YAP/TAZ dKO LLC cells (Fig. EV4A). To elucidate the mechanisms by which YAP/TAZ dKO cells protect their surrounding WT cells from ferroptosis, we characterized the transcriptome of these cells using RNA sequencing (RNA-seq) analysis. The RNA-seq data are visualized in a volcano plot (Fig. 3A), which highlights the genes exhibiting significant differential expression between YAP/TAZ dKO and WT cells. This analysis revealed 910 differentially expressed genes (DEGs), of which 598 were upregulated (Dataset EV1), and 312 were downregulated (Dataset EV2) in YAP/TAZ dKO cells relative to WT cells, marked by a fold change threshold of >2 and an adjusted P value of < 0.01. Of these DEGs, we focused on ferroptosis-related genes (Agrawal et al, 2024) (Table EV2) and found that solute carrier family 7 member 11 (Slc7a11) and GTP cyclohydrolase 1 (Gch1) showed significant alterations in their gene expression levels (Fig. 3A). Given their established roles in protection against ferroptosis (Jiang et al, 2021), we focused on these genes for further validation. Of note, the log2 fold change of GCH1 was higher than that of SLC7A11 (also known as xCT) (5.26 and 2.06, respectively) (Dataset EV1). Moreover, despite the increased mRNA levels of SLC7A11, intracellular glutathione (GSH) levels were comparable between WT and YAP/TAZ dKO cells (Fig. EV4B), suggesting that elevated mRNA expression of SLC7A11 may not functionally contribute to GSH redox homeostasis and ferroptosis resistance in YAP/TAZ dKO cells. In contrast, GCH1 protein levels showed a marked increase in YAP/TAZ dKO cells compared to WT cells (Fig. 3B), likely reflecting the difference in their relative mRNA expression levels (Fig. 3A). Indeed, reverse transcription and real-time PCR analysis confirmed that the Gch1 mRNA levels were dramatically increased in YAP/TAZ dKO cells compared to those in WT cells (Fig. 3C). We also found that the mRNA abundance of Gch1 was higher in YAP/TAZ-deficient subcutaneous tumors than in WT tumors (Fig. 3D), indicating that YAP/TAZ negatively regulated GCH1 expression in vivo. Further verifying the key role of YAP/TAZ in regulating GCH1, YAP re-expression suppressed GCH1 expression at the protein (Fig. 3E) and mRNA levels (Fig. 3F). Moreover, inactivating endogenous YAP/TAZ through serum starvation (Yu et al, 2012) or high-cell-density culture (Zhao et al, 2007) induced Gch1 expression (Fig. 3G), while inhibiting its transcriptional activity (Fig. EV4C). Furthermore, GCH1 expression was negatively correlated with both YAP expression (Fig. EV4D) and the previously characterized YAP/TAZ transcriptional target signature of 22 genes (Wang et al, 2018) across multiple human tissues (Fig. EV4E). These results suggest that although YAP/TAZ generally function as transcriptional co-activators, they suppress transcription at the Gch1 locus, thereby reducing Gch1 mRNA expression. Indeed, assay for transposase-accessible chromatin with sequencing (ATAC-seq) analysis revealed that peaks indicative of open chromatin were observed at approximately +1 kb of the transcription start site of the Gch1 locus in YAP/TAZ dKO cells compared to those in WT cells (Fig. 3H). Together, our data indicate that YAP/TAZ contribute to maintaining a closed chromatin state in the regulatory regions of Gch1, thereby suppressing Gch1 transcription.

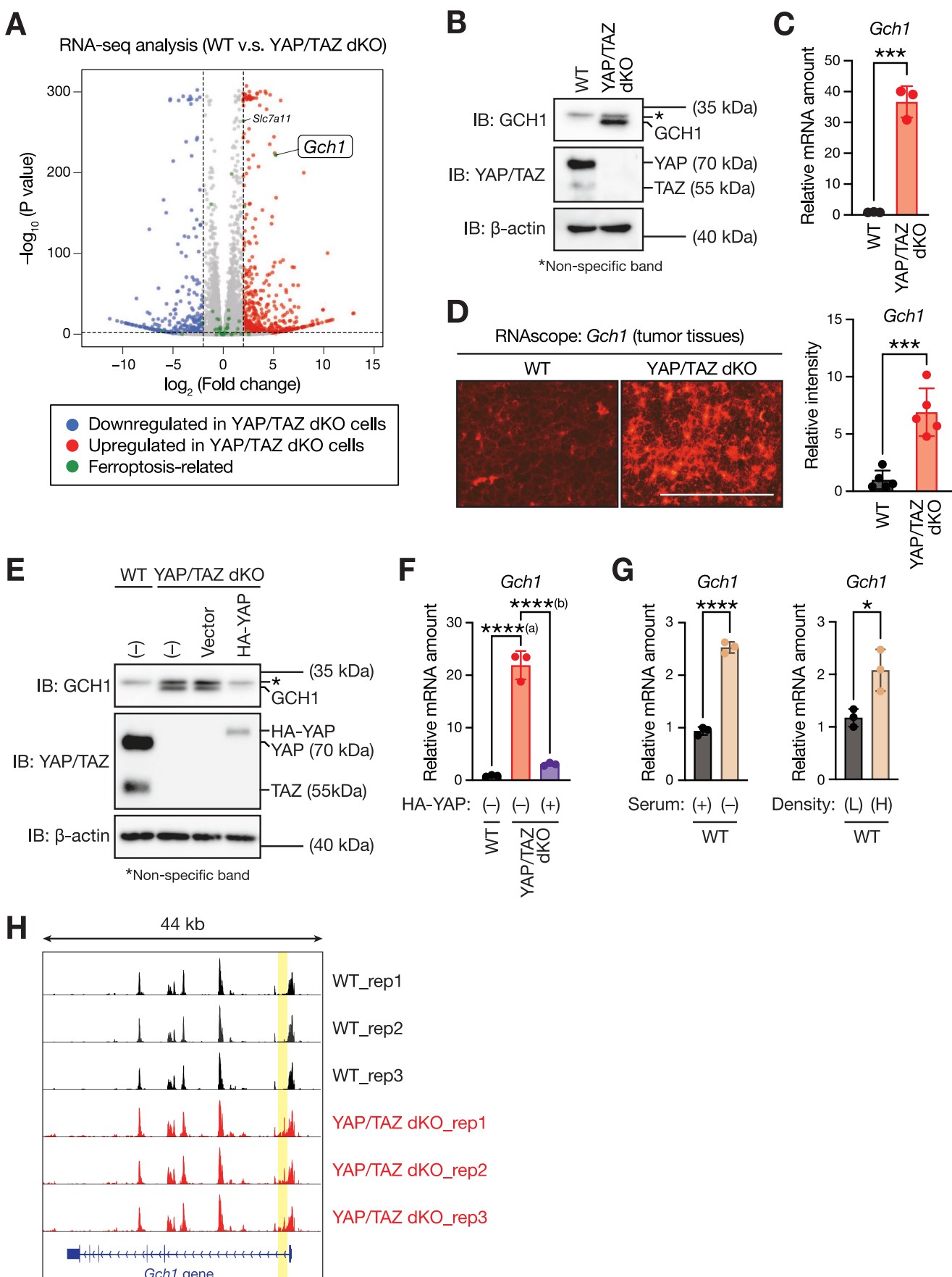

◄ **Figure 3. YAP/TAZ loss induces *Gch1* transcription.**

(A) Differences in the transcriptomes of WT and YAP/TAZ dKO LLC cells. Differentially expressed genes (DEGs; fold change >2 and adjusted *P* value < 0.01) in YAP/TAZ dKO LLC cells compared to WT cells are displayed in a volcano plot (blue, downregulated; red, upregulated). The ferroptosis-related genes are highlighted in green. Data represent the mean of three biologically independent samples from a representative experiment. Statistical analysis of differential gene expression was performed using DESeq2. (B) GCH1 accumulates in YAP/TAZ-deficient LLC cells. Immunoblotting (IB) of cell extracts from WT and YAP/TAZ dKO LLC cells with antibodies against indicated proteins was shown. (C) Loss of YAP/TAZ induces *Gch1* transcription. Total RNA extracted from WT or YAP/TAZ dKO LLC cells were subjected to reverse transcription (RT) and real-time PCR analysis. Data are means ± SD of three biologically independent samples from a representative experiment.
***$P$ = 0.000253798986094 (unpaired *t* test). (D) *Gch1* mRNA abundance is increased in YAP/TAZ-deficient lung tumor tissues. Equal numbers of WT or YAP/TAZ dKO LLC cells were subcutaneously injected into C57BL/6 mice. Fourteen days after transplantation, tumors were harvested and then subjected to RNAscope analysis of *Gch1* mRNA. Quantitative analysis of signal *Gch1* intensity is shown on the right. Data are means ± SD of five randomly selected images. Scale bars, 100 μm.
***$P$ = 0.000361619614793 (unpaired *t* test). (E) YAP re-expression prevents GCH1 accumulation in YAP/TAZ-deficient LLC cells. WT and YAP/TAZ dKO LLC cells infected (or not) with expression vectors for HA-YAP were subjected to immunoblot (IB) analysis of the indicated proteins. (F) Re-expression of YAP suppresses *Gch1* transcription in YAP/TAZ dKO LLC cells. Total RNA extracted from the indicated cells were subjected to RT and real-time PCR analysis. Data are means ± SD of three biologically independent samples from a representative experiment. ****[a]$P$ = 0.000000309035446; ****[b]$P$ = 0.000000796405245 (one-way ANOVA test followed by Tukey's multiple comparison test). (G) YAP/TAZ inhibition conditions induce *Gch1* transcription. WT LLC cells were cultured with or without serum for 32 h (left) or at low (L) or high (H) cell density for 52 h (right), and then subjected to RT and real-time PCR analysis. Data are means ± SD of three biologically independent samples from a representative experiment. *$P$ = 0.021959987914851; ****$P$ = 0.000029784548704 (unpaired *t* test). (H) YAP/TAZ loss affects chromatin accessibility of *Gch1*. WT and YAP/TAZ dKO LLC cells were subjected to assay for transposase-accessible chromatin with sequencing (ATAC-seq). Differential peaks indicative of the open chromatin status are highlighted in yellow. Source data are available online for this figure.

## YAP/TAZ loss increases tetrahydrobiopterin production to protect neighboring cells from ferroptosis

GCH1 is a rate-limiting enzyme in the biosynthesis of tetrahydrobiopterin (BH4), a potent radical-trapping antioxidant that prevents lipid peroxidation (Soula et al, 2020). Lipid peroxide is eliminated by BH4 to BH2 (dihydrobiopterin) oxidation. Inactive BH2 is then reduced by dihydrofolate reductase (DHFR) to regenerate active BH4 (Kraft et al, 2020) (Fig. 4A). Because YAP/TAZ loss induced GCH1 expression in LLC cells, the biopterin levels were measured using liquid chromatography-electrospray ionization-mass spectrometry (LC-MS) in these cells. LC-MS analysis revealed that the intracellular BH4 (Fig. 4B) and BH2 levels (Fig. EV5A) were markedly higher in YAP/TAZ dKO cells than in WT cells. BH4 is a secretory metabolite that affects surrounding cells (Walter et al, 1994). Indeed, levels of BH2, the oxidized form of BH4, were markedly higher in the culture medium of YAP/TAZ dKO cells than in those of WT cells, suggesting an increased secretion of BH4/BH2 in YAP/TAZ dKO cells (Fig. EV5B). These observations led us to hypothesize that elevated BH4 production in YAP/TAZ dKO cells may contribute to the ferroptosis resistance of YAP/TAZ dKO cells and protect neighboring WT cells from ferroptosis. Therefore, YAP/TAZ/GCH1 triple-KO (tKO) cells were generated using the CRISPR/Cas9 system to test this hypothesis. GCH1 deletion was confirmed using DNA sequencing (Fig. EV5C) and immunoblotting (Fig. 4C). Additional deletion of GCH1 in YAP/TAZ dKO cells significantly suppressed BH4 (Fig. 4D) and BH2 production (Fig. EV5D), confirming the functional loss of GCH1 in YAP/TAZ/GCH1 tKO cells. Notably, the additional loss of GCH1 in YAP/TAZ dKO cells resensitized these cells to ferroptosis, which was prevented by adding BH4 to the culture medium (Fig. 4E). Consistent with this observation, the cellular lipid peroxidation levels were inversely correlated (Fig. EV5E). These results suggest that GCH1 induction and BH4 accumulation are the major contributors to the acquired ferroptosis resistance in YAP/TAZ dKO LLC cells. More importantly, while RSL3-induced ferroptosis was largely suppressed in WT cells co-cultured with YAP/TAZ dKO cells (Fig. 2C), YAP/TAZ/GCH1 tKO cells failed to protect neighboring WT cells from

ferroptosis (Fig. 4F). These results suggest that GCH1 is required for YAP/TAZ dKO cells to protect surrounding cells from ferroptosis. Collectively, our observations suggest that the YAP/TAZ–GCH1–BH4 axis is integral for LLC cells to confer resistance to ferroptosis in both cell-autonomous and non-cell-autonomous manners.

## GCH1 inhibitor sensitizes tumors to the ferroptosis-inducing drug

Given that the GCH1–BH4 axis is crucial for inducing ferroptosis resistance in YAP/TAZ dKO cells, we investigated whether pharmacological inhibition of GCH1 could sensitize these cells to RSL3-induced ferroptosis. To this end, we utilized 2,4-Diamino-6-hydroxypyrimidine (DAHP) to inhibit GCH1 enzymatic activity (Kolinsky and Gross, 2004). Neither RSL3 nor DAHP alone induced cell death in YAP/TAZ dKO LLC cells; however, combining them induced cell death in YAP/TAZ dKO cells, which was prevented by pretreatment with ferrostatin-1 or BH4 (Fig. 5A). These results suggest that DAHP treatment inhibits the GCH1–BH4 axis in YAP/TAZ dKO cells, thereby sensitizing them to RSL3-induced ferroptosis.

Given that cancer cells exhibit a heterogeneous activation status of YAP/TAZ in vivo (Fig. 1A and Fig. 2B), and that YAP/TAZ inhibition within cancer cells confers multicellular-level protection against ferroptosis (Fig. 2C), we hypothesized that heterogeneous YAP/TAZ activity within tumor tissues may contribute to tumor resistance to ferroptosis. Therefore, targeting the GCH1–BH4 axis in combination with ferroptosis inducers could be a potential therapeutic approach for tumors exhibiting heterogeneous YAP/TAZ activity. To test this hypothesis, LLC cells were transplanted into immunocompetent syngeneic mice and the therapeutic efficacy of this combined approach was evaluated (Fig. 5B). While treatment with RSL3 or DAHP alone did not affect LLC tumor growth, combining the two drugs significantly suppressed LLC tumor growth (Fig. 5C). Consistently, the tumor weight 18 days after transplantation was reduced by approximately 65% by the combination of RSL3 and DAHP (Fig. 5D). Additionally, combined treatment with RSL3 and DAHP significantly prolonged the

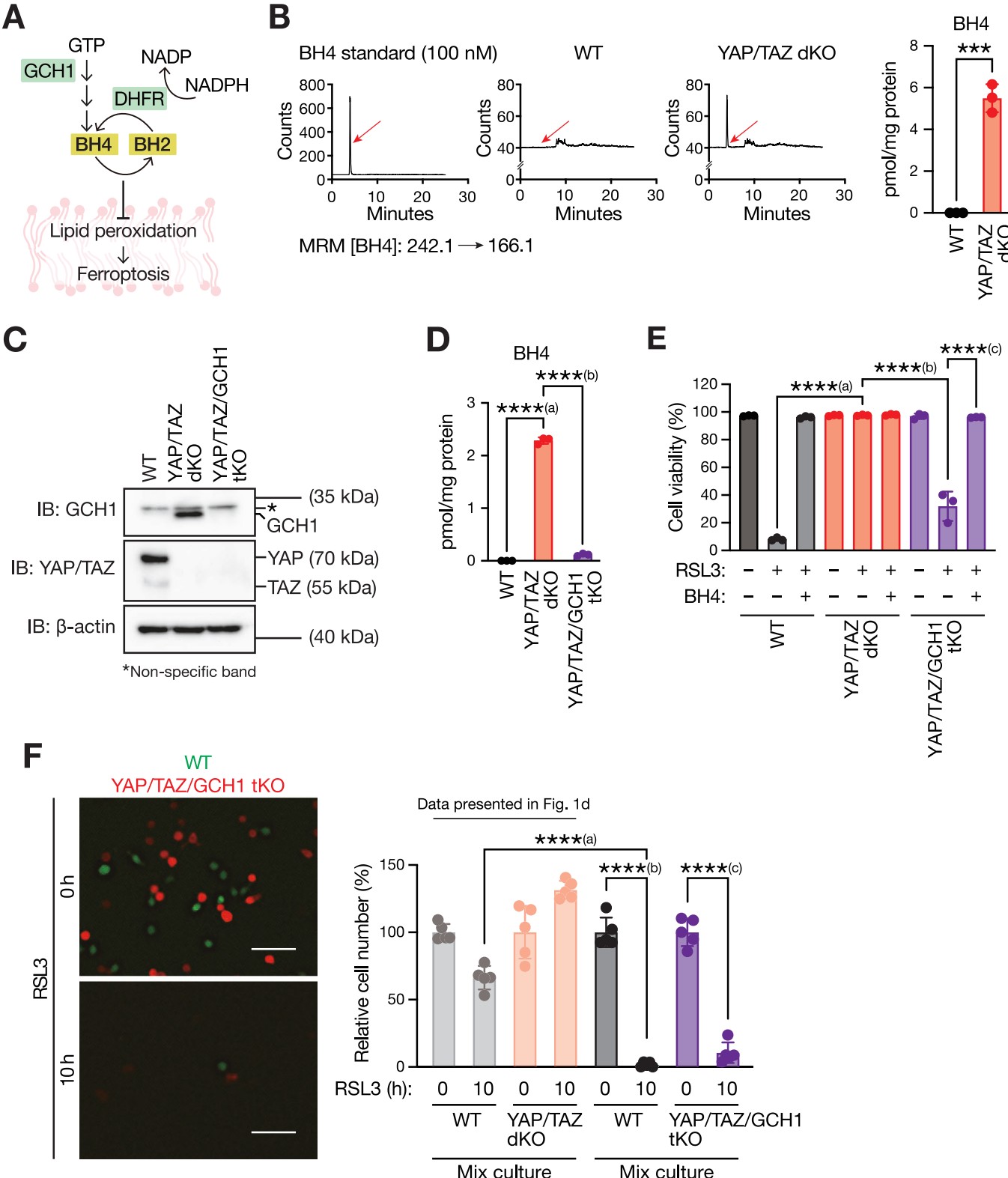

**Figure 4. Loss of YAP/TAZ increases tetrahydrobiopterin production to protect neighboring cells from ferroptosis.**

(A) Schematic representation of the tetrahydrobiopterin biosynthesis pathway. GCH1 GTP cyclohydrolase 1, DHFR dihydrofolate reductase, BH4 tetrahydrobiopterin, BH2 dihydrobiopterin. (B) YAP/TAZ loss increases BH4 production. Peaks with the corresponding multiple reaction monitoring (MRM) indicate intracellular BH4 levels (left, red arrows). Quantitative analysis of the BH4 levels is shown on the right-hand side. Data are means ± SD of three biologically independent samples from a representative experiment. ***$P$ = 0.000144582491651 (unpaired $t$ test). (C) Generating YAP/TAZ/GCH1 triple-knockout (tKO) cells. Immunoblot (IB) analysis of cell extracts from WT, YAP/TAZ dKO, and YAP/TAZ/GCH1 tKO LLC cells with antibodies to the indicated proteins was shown. (D) GCH1 is required for BH4 accumulation in YAP/TAZ dKO cells. The levels of BH4 in WT, YAP/TAZ dKO, and YAP/TAZ/GCH1 tKO LLC cells were measured by LC-MS/MS. Data are means ± SD of three biologically independent samples from a representative experiment. ****[a]$P$ = 0.000000000001766; ****[b]$P$ = 0.000000000006759 (one-way ANOVA test followed by Tukey's multiple comparison test). (E) GCH1 is required for ferroptosis resistance in YAP/TAZ-deficient cells, and BH4 supplementation confers such resistance. WT, YAP/TAZ dKO and YAP/TAZ/GCH1 tKO LLC cells were pretreated (or not) with BH4 (40 μM) for 30 min, followed by stimulation with RSL3 (400 nM) for 10 h. Dead cells were stained with propidium iodide (PI), and the percentage of the PI-negative live cell population was calculated using the flow cytometer. Data are means ± SD of three biologically independent samples from a representative experiment. ****[a]$P$ = 0.000000000084054; ****[b]$P$ = 0.000000005054431; ****[c]$P$ = 0.000000006325909 (one-way ANOVA test followed by Tukey's multiple comparison test). (F) YAP/TAZ-deficient cells lacking GCH1 expression fail to protect surrounding cells from ferroptosis. WT LLC cells expressing EGFP (green) were co-cultured with YAP/TAZ/GCH1 tKO LLC cells expressing tdTomato (red) in the presence of RSL3 (400 nM) for 10 h. Quantitative analysis of cell numbers is shown on the right. Data are means ± SD of five randomly selected images. The experiment was repeated three times, and data from one representative experiment are shown. The relative number of WT LLC cells co-cultured with YAP/TAZ dKO LLC cells in Fig. 1d is shown in a lighter color for reference. Scale bars, 100 μm. ****[a]$P$ = 0.000000000597045; ****[b]$P$ = 0.000000000000105; ****[c]$P$ = 0.000000000000219 (one-way ANOVA test followed by Tukey's multiple comparison test). Source data are available online for this figure.

survival of tumor-bearing mice compared with that of vehicle-treated mice (Fig. 5E). These results suggest that pharmacologically inhibiting the GCH1–BH4 axis in tumors exhibiting YAP/TAZ heterogeneity prevents cancer cells from developing resistance to RSL3-induced ferroptosis. Therefore, this combination approach potentiates the efficacy of ferroptosis inducers, leading to tumor destruction through ferroptosis.

## Discussion

In this study, we demonstrated the tumor-protective role of intratumoral heterogeneity in the Hippo pathway in conferring ferroptosis resistance (Fig. 6). YAP/TAZ expression patterns are heterogeneous among cancer cells in invasive lung adenocarcinoma, which is associated with poor prognosis. The syngeneic model of murine lung cancer, LLC, recapitulates the heterogeneous YAP/TAZ activation status within tumor tissues. Through a series of in vitro experiments, we showed that LLC cells with low YAP/TAZ activity exhibited slow growth but were resistant to ferroptosis. In contrast, cells with high YAP/TAZ activity had higher cell proliferation but were sensitive to ferroptosis. These findings highlight that cellular differences in YAP/TAZ activity are directly linked to their ferroptosis sensitivity, establishing a functional consequence of this heterogeneity within tumors. Importantly, ferroptosis resistance acquired by cancer cells with low YAP/TAZ activity propagated to adjacent cancer cells, conferring multicellular protection against ferroptosis in tumors. Consequently, pharmacologically inhibiting this transmission sensitized LLC tumors to ferroptosis in vivo, leading to tumor destruction. Thus, our results suggest that the intratumoral heterogeneity of the Hippo pathway establishes a symbiotic relationship to overcome lipid peroxidative stress within the tumor microenvironment. A critical implication of our findings is that studying YAP/TAZ heterogeneity requires single-cell resolution. Bulk RNA sequencing, which captures average gene expression across entire tumor tissues, obscures the cell-to-cell variation that underlies functional differences in ferroptosis sensitivity. For example, tumors with the same average YAP/TAZ expression may have profoundly different cellular compositions—ranging

from uniformly moderate YAP/TAZ activity to a mixture of YAP/TAZ[high] and YAP/TAZ[low] cells. Our study indicates that this heterogeneity itself has biological significance, particularly in shaping non-cell-autonomous resistance to ferroptosis. Therefore, while publicly available bulk datasets such as TCGA are valuable for many types of analyses, they are not suitable for investigating the functional consequences of YAP/TAZ heterogeneity. Single-cell transcriptomic or spatial profiling technologies are required to uncover how Hippo pathway heterogeneity contributes to tumor adaptation and therapeutic resistance.

The Hippo pathway was initially identified in flies through genetic mosaic screening as a tumor-suppressive pathway that restricts cell proliferation. The discovery that this pathway, along with its role in cell proliferation, is conserved in mammals has generated considerable interest, driving the identification of its significant role in various aspects of cancer progression (Baroja et al, 2024; Franklin et al, 2023; Piccolo et al, 2023). Previous studies have convincingly established the cell-autonomous functions of the Hippo pathway in the regulation of cancer development, growth, drug resistance, and metastasis. Additionally, the complexity of the Hippo pathway has greatly increased with the discovery of non-cell-autonomous functions. For instance, although YAP/TAZ are generally considered to promote tumor progression in a cell-autonomous manner, peritumoral activation of YAP/TAZ in normal hepatocytes triggers regression of primary liver tumors and melanoma-derived liver metastases through a mechanism involving cell competition (Moya et al, 2019). These emerging roles of the Hippo pathway in multicellular organisms, wherein different cell types coordinately maintain tissue homeostasis by interacting with each other, may help elucidate the context-dependent functions of this pathway in cancer progression. In the present study, we demonstrated that while cancer cells with high YAP/TAZ activity have a competitive growth advantage, they are sensitive to lipid peroxide stress. To overcome this limitation, surrounding cancer cells with low YAP/TAZ activity provide BH4 to create a protective lipid peroxide stress-free microenvironment. Although YAP/TAZ[low] cells may not primarily contribute to tumor growth because of their growth disadvantage compared to YAP/TAZ[high] cells, they help construct an ideal microenvironment against ferroptosis, thereby contributing to drug resistance. This

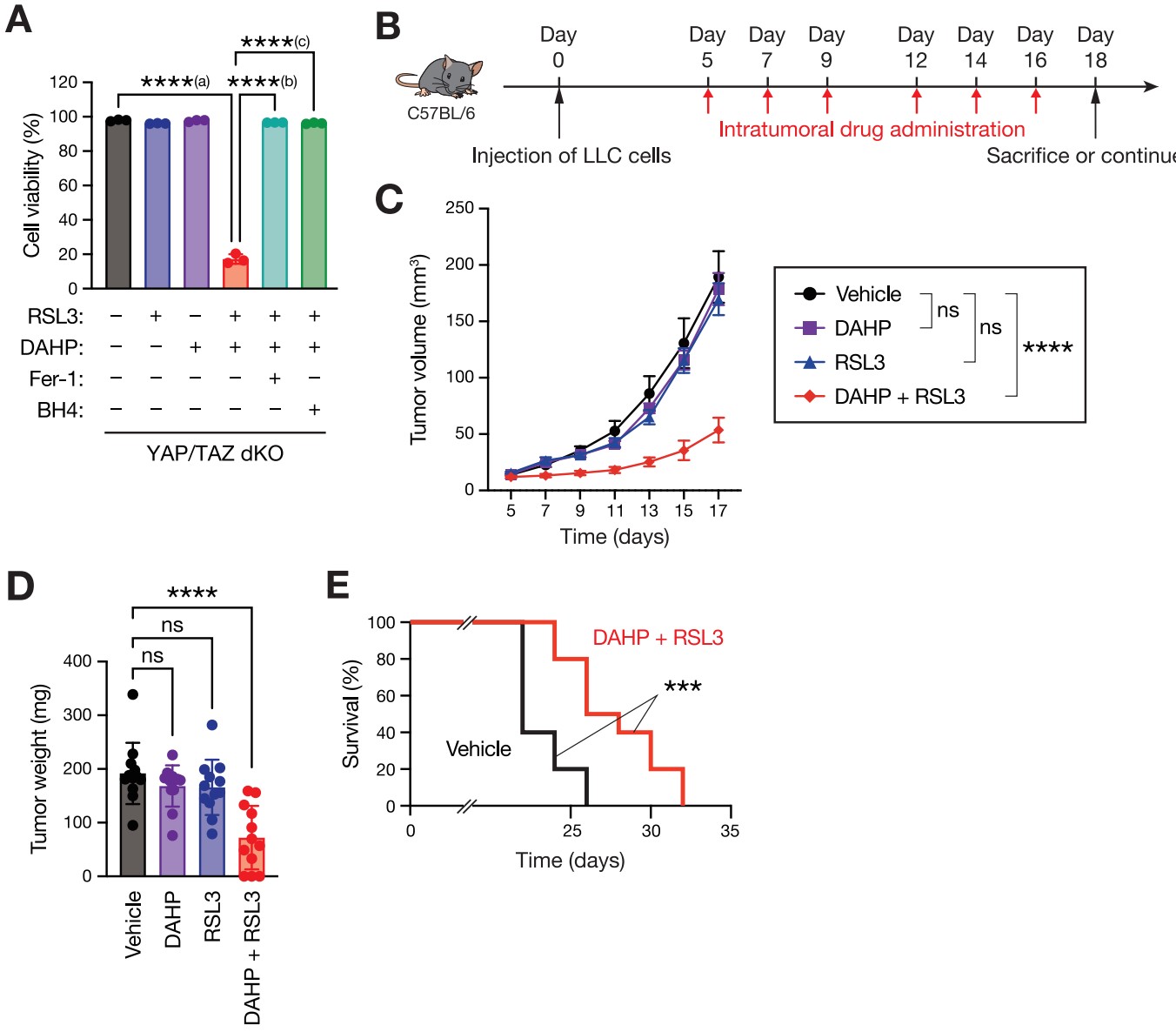

**Figure 5. GCH1 inhibitor sensitizes tumors to the ferroptosis-inducing drug.**

(A) Pharmacological inhibition of GCH1 using 2,4-diamino-6-hydroxypyrimidine (DAHP) sensitizes YAP/TAZ-deficient cells to ferroptosis. YAP/TAZ double-knockout (dKO) LLC cells were pretreated (or not) with Ferrostatin-1 (Fer-1; 5 μM) or BH4 (40 μM) for 30 min, and then stimulated with RSL3 (400 nM) and/or DAHP (10 mM) for 24 h. The percentage of live cell population was calculated using PI and flow cytometer. Data are means ± SD of three biologically independent samples from a representative experiment. ****(a)$P = 0.000000000000026$; ****(b)$P = 0.000000000000026$; ****(c)$P = 0.000000000000026$ (one-way ANOVA test followed by Tukey's multiple comparison test). (B) Schematic diagram of experimental design. Wild-type LLC lung cancer cells were subcutaneously injected into C57BL/6 mice, and the drug administration was started at day 5. Tumor-bearing mice were treated with intratumoral injections of vehicle (10% DMSO, 45% PEG300, and 45% PBS), DAHP (100 mg/kg body weight in the vehicle), or RSL3 (50 mg/kg body weight in the vehicle) three times a week for the whole duration of the experiment, until a mouse was sacrificed. (C) Administering the GCH1 inhibitor DAHP sensitizes LLC tumors to RSL3. LLC cells were transplanted into C57BL/6 mice, and tumor growth over time following indicated treatments was monitored. Data are presented as the means ± SEM; $n = 12$ tumors for each group. ns, not significant ($P = 0.975133563807377$ for Vehicle v.s. DAHP, $P = 0.873296292458696$ for Vehicle v.s. RSL3); ****$P = 0.000000000479864$ (two-way ANOVA test). (D) The RSL3 and DAHP combination suppressed LLC tumor growth. C57BL/6 mice were injected with LLC cells, and tumor weight was determined after 2 weeks of drug treatment. Data are presented as the means ± SD; $n = 12$ tumors for each group. ns, not significant ($P = 0.689283972138939$ for Vehicle v.s. DAHP, $P = 0.617118340941646$ for Vehicle v.s. RSL3); ****$P = 0.000007129017268$ (one-way ANOVA test followed by Tukey's multiple comparison test). (E) Combination RSL3 with DAHP therapy improves the survival of mice bearing LLC tumors. Kaplan–Meier tumor-free survival curves for tumor-bearing mice treated with the indicated drugs were shown. $n = 10$ mice per group. ***$P = 0.000996737657140$ (log-rank test). Source data are available online for this figure.

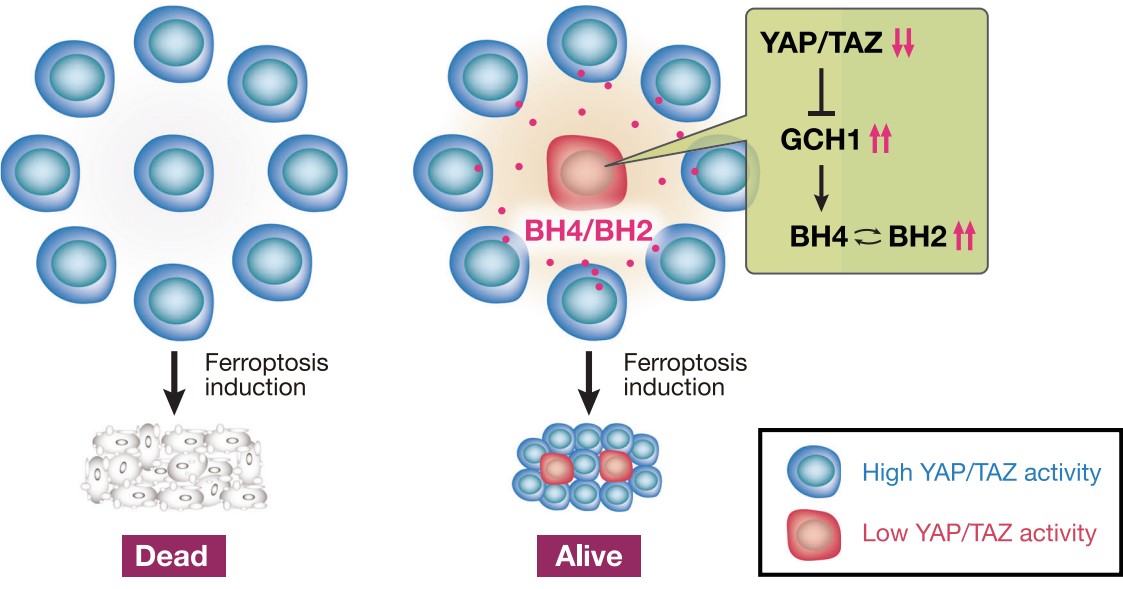

**Figure 6.  Proposed model for ferroptosis resistance conferred by YAP/TAZ heterogeneity in lung cancer.**

In tumors with heterogeneous YAP/TAZ activity, YAP/TAZ^low cancer cells exhibit high GCH1 expression, thereby producing large amounts of BH4 and BH2 to protect both themselves and neighboring cells from ferroptosis.

study sheds light on the symbiotic relationship between cancer cells with different Hippo pathway statuses during tumor growth and progression.

Cancer cell ferroptosis has recently gained attention as a potential novel treatment strategy. Numerous studies have investigated the therapeutic effects of drugs targeting ferroptosis-related molecules such as GPX4 and xCT on various types of cancers. In addition to the therapeutic induction of ferroptosis, accumulating evidence has suggested that cancer cells encounter pro-ferroptotic stimuli within the tumor microenvironment or during metastasis. For instance, $CD8^+$ T cells suppress xCT expression in cancer cells via IFNγ, thereby sensitizing them to ferroptosis (Wang et al, 2019). Another study proposed that metastasizing melanoma cells are subjected to more ferroptosis stress in the blood than in the lymph, which may explain why metastasis through the blood is less efficient compared to lymphatic metastasis (Ubellacker et al, 2020). Previous studies have revealed the cell-autonomous functions of the Hippo pathway in regulating cellular ferroptosis sensitivity. YAP promotes ferroptosis in color-ectal cancer by transcriptionally upregulating acyl-CoA synthetase long-chain family member 4 (ACSL4) and transferrin receptor (TFRC) expression (Wu et al, 2019). TAZ induces ferroptosis in renal cell carcinoma by upregulating epithelial membrane protein 1 (EMP1) expression, which promotes NADPH oxidase 4 (NOX4) (Yang et al, 2019). Furthermore, TAZ also upregulates NOX2 through angiopoietin-like 4 (ANGPTL4), a direct target gene of TAZ, thereby increasing susceptibility to ferroptosis in ovarian cancer cells (Yang et al, 2020). All of these studies have revealed the ferroptosis-promoting functions of YAP/TAZ in different cancer cell types. However, no characterized mechanisms explain the non-cell-autonomous effects of the Hippo pathway in protecting neighboring cells against ferroptosis. In this study, we demonstrated that YAP/TAZ negatively regulated the expression of

GCH1, a key enzyme for biopterin metabolism, thereby controlling the production of a secretory antioxidant metabolite BH4. Additional deletion of GCH1 in YAP/TAZ-deficient LLC cells not only sensitized these cells to ferroptosis but also prevented the propagation of ferroptosis resistance between cancer cells. Our experimental evidence indicated that pharmacological inhibition of GCH1 using DAHP, in combination with the ferroptosis inducer RSL3, efficiently suppressed LLC tumor growth and prolonged the survival of tumor-bearing mice. Therefore, the YAP/TAZ–GCH1–BH4 axis is integral to establishing multicellular-level protection against ferroptosis in lung cancer. Limitations of the study include the unclear mechanism of GCH1 upregulation upon YAP/TAZ loss. Additionally, future studies using clinical samples are needed to validate the relationship between YAP/TAZ heterogeneity and GCH1 in human cancer.

In addition to its role in protecting against ferroptosis, BH4 is an essential cofactor for several enzymes, including aromatic amino acid hydroxylases, alkylglycerol monooxygenase, and nitric oxide synthases (Kim and Han, 2020; Werner et al, 2011). Therefore, BH4 is found in most tissues and plays an important role in various pathological conditions, including neurodegenerative and neurodevelopmental disorders, cardiovascular and endothelial dysfunction, inflammation, and pain sensitivity. Future studies investigating the role of the YAP/TAZ–GCH1–BH4 axis in these contexts, along with the elucidation of mechanisms underlying the cellular uptake of BH2/BH4 (Ohashi et al, 2017; Ohashi et al, 2011), may provide biological insights into the diverse roles of the Hippo pathway in multicellular organisms. Therefore, this study provides important implications that the GCH1–BH4 axis is a key downstream element of the Hippo pathway in some physiological and pathological contexts.

Recent studies have suggested the role of YAP/TAZ hetero-geneity in mammalian development (Nita and Moroishi, 2024). At the mid-blastocyst stage, there is considerable variability in nuclear

YAP activity and the expression of pluripotency factors within the inner cell mass. As epiblast formation progresses, YAP gradually accumulates in the nucleus, triggering the activation of TEAD-driven transcriptional programs. The increased TEAD transcriptional activity promotes the expression of pluripotency factors, which are crucial for epiblast formation. During this process, cells compete to eliminate those with low TEAD activity via apoptosis, effectively removing less pluripotent cells. This mechanism ensures the generation of high-quality epiblasts, which are characterized by a state of naïve pluripotency (Hashimoto and Sasaki, 2020). Given the recently reported physiological role of ferroptosis in avian muscle tissue sculpting (Co et al, 2024), it would be an intriguing future direction to explore whether ferroptosis regulation by YAP/TAZ heterogeneity contributes to organ development. Furthermore, identifying the molecular mechanisms underlying the development of YAP/TAZ heterogeneity in both physiology and pathology is an important avenue for future research.

# Methods

### Reagents and tools table

| Reagent/resource | Reference or source | Identifier or catalog number |
|---|---|---|
| **Experimental models** | | |
| Mouse LLC cells | ATCC | CRL-1642 |
| Human LK-2 | RIKEN BRC | RCB1970 |
| Human PC-9 | RIKEN BRC | RCB4455 |
| Human A549 | ATCC | CCL-185 |
| Human H1975 | ATCC | CRL-5908 |
| C57BL/6J mice | Kyudo | N/A |
| **Recombinant DNA** | | |
| Plasmid: PX459 (CRISPR) | Addgene | Plasmid #62988 |
| Plasmid: lentiCRISPR v2 (CRISPR) Puro | Derived from Addgene plasmid #83480 by replacing the blasticidin S-resistance gene with a puromycin-resistance gene | Plasmid # 83480 |
| pLVSIN-CMV vector | Takara | Cat#: 6183 |
| pLVSIN-CMV HA-YAP | This paper | N/A |
| pLVSIN-CMV EGFP | This paper | N/A |
| pLVSIN-CMV tdTomato | This paper | N/A |
| pLenti-P2A and pLenti-P2B | Cosmobio | Cat#: LV003 |
| **Antibodies** | | |
| Anti-YAP/TAZ antibody | Santa Cruz | Cat#: sc-101199; RRID: AB_1131430 |
| Anti-cytokeratin (CK; clone AE1/AE3) antibody | Agilent Technologies | Cat#: M3515 |
| Anti-GCH1 antibody | Thermo Fisher Scientific | Cat#: PA5-120098; RRID: AB_2913670 |
| Anti-YAP/TAZ antibody | Cell Signaling | Cat#: 93622 |

| Reagent/resource | Reference or source | Identifier or catalog number |
|---|---|---|
| Anti-HA-Tag antibody | Cell Signaling | Cat#: 2999 |
| Anti-β-actin antibody | Cell Signaling | Cat#: 12620 |
| Horseradish peroxidase (HRP)-linked mouse IgG | Cell Signaling | Cat#: 7076 |
| Horseradish peroxidase (HRP)-linked rabbit IgG | Cell Signaling | Cat#: 7074 |
| **Oligonucleotides and other sequence-based reagents** | | |
| CRISPR targeting sequence: human YAP: 5′-CATCAG ATCGTGCACGTCCG-3′ | This paper | N/A |
| CRISPR targeting sequence: human TAZ: 5′-TGTCTAGG TCCTGCGTGACG-3′ | This paper | N/A |
| CRISPR targeting sequence: mouse *Yap*: 5′-GGCAGCTT GCGAAGCCGCAT-3′ | This paper | N/A |
| CRISPR targeting sequence: mouse *Taz*: 5′-GGATTGGCGCG AGTGCGAGC-3′ | This paper | N/A |
| CRISPR targeting sequence: mouse *Gch1*: 5′- GCTCGGAGA ACCCATTGGTG-3′ | This paper | N/A |
| qPCR primers for mouse *Gapdh*: 5′-GCCTGGAG AAACCTGCCAAGTATG-3′ and 5′-GAGTGGGGAGTTGCTG TTGAAGTCG-3′ | This paper | N/A |
| qPCR primers for mouse *Ankrd1*: 5′-CGATGAGTATAAACGGAC GGCACTC-3′ and 5′-GGATG GCTGTGGATTCAAGCATA TC-3′ | This paper | N/A |
| qPCR primers for mouse *Gch1*: 5′-AGCAAGTCCTTGGTCTC AGTAAAC-3′ and 5′-ACC GCAATCTGTTTGGTGAGGC-3′ | This paper | N/A |
| qPCR primers for mouse *Tfrc*: 5′-GAAGTCCAGTGTGGG AACAGGT-3′ and 5′-CAAC CACTCAGTGGCACCAACA-3′ | This paper | N/A |
| qPCR primers for mouse *Emp1*: 5′-TCCCTGTCCTACGGCAA TGAAG-3′ and 5′-CTGGA CACGAAGACCACAAGG-3′ | This paper | N/A |
| qPCR primers for mouse *Acsl4*: 5′-CCTTTGGCTCATGTGCT GGAAC-3′ and 5′-GCCAT AAGTGTGGGTTTCAGTAC-3′ | This paper | N/A |
| qPCR primers for mouse *Angptl4*: 5′-CTGGACAGTGATTCAGAGA CGC-3′ and 5′-GATGCTGTGCA TCTTTTCCAGGC-3′ | This paper | N/A |
| **Chemicals, enzymes and other reagents** | | |
| (1S,3 R)-RSL3 | Selleck Chemicals | Cat#: S8155 |
| Erastin | Cayman | Cat#: 17754 |
| L-Buthionine-(S,R)-Sulfoximine | Cayman | Cat#: 14484 |
| Sulfasalazine | MedChemExpress | Cat#: HY-14655 |
| RPMI 1640 | MP Biomedicals | Cat#: 1646454 |

| Reagent/resource | Reference or source | Identifier or catalog number |
|---|---|---|
| Dialyzed Fetal Bovine Serum | Serana | Cat#: S-FBS-NL-065 |
| L-Methionine | Nacalai | Cat#: 13038-72 |
| L-Glutamine | FUJIFILM Wako | Cat#: 073-05391 |
| actinomycin D | FUJIFILM Wako | Cat#: 018-21264 |
| TNF-α | Abcam | Cat#: ab206641 |
| cycloheximide | FUJIFILM Wako | Cat#: 037-20991 |
| Z-VAD-FMK | Selleck Chemicals | Cat#: S7023 |
| Necrostatin-1 | Selleck Chemicals | Cat#: S8037 |
| Ferrostatin-1 | Cayman Chemical | Cat#: 17729 |
| (6 R)-5,6,7,8-tetrahydro-L-biopterin | Cayman Chemical | Cat#: 81880 |
| 7,8-dihydro-L-biopterin | Cayman Chemical | Cat#: 81882 |
| DAHP | Cayman Chemical | Cat#: 81260 |
| Lipofectamine 3000 transfection kit | Invitrogen | Cat#: L3000-08 |
| Puromycin | Thermo Fisher Scientific | Cat#: A1113803 |
| PolyJet | SignaGen | Cat#: SL100688 |
| Polybrene | Merck | Cat#: TR-1003-G |
| Propidium iodide | Dojindo | Cat#: 341-07881 |
| BODIPY 581/591 C11 | Thermo Fisher Scientific | Cat#: D3861 |
| Phos-tag (TM) Acrylamide AAL-107 | FUJIFILM Wako | Cat#: 304-93521 |
| **Software** | | |
| Prism 10 | GraphPad | https://www.graphpad.com/ |
| FlowJo | BD Biosciences | https://flowjo.com/solutions/flowjo/downloads |
| Adobe Illustrator | Adobe | https://www.adobe.com/products/illustrator.html |
| Adobe Photoshop | Adobe | https://www.adobe.com/products/illustrator.html |
| **Other** | | |
| Peroxidase Stain DAB Kit | Nacalai Tesque | Cat#: 25985-50 |
| 3-amino-9-ethylcarbazole | Nichirei | Cat#: 415182 |
| VECTASTAIN Elite ABC Rabbit IgG Kit | Vector Laboratories | Cat#: PK-6101 |
| TRIzol reagent | Invitrogen | Cat#: 15596026 |
| ReverTra Ace qPCR RT kit | TOYOBO | Cat#: FSQ-301 |
| SYBR qPCR Mix | TOYOBO | Cat#: QPX-201 |
| RNeasy Mini Kit | QIAGEN | Cat#: 74104 |
| NEBNext Poly(A) mRNA Magnetic Isolation Module | New England Biolabs | Cat#: E7490 |

| Reagent/resource | Reference or source | Identifier or catalog number |
|---|---|---|
| NEBNext Ultra II Directional RNA Library Prep Kit | New England Biolabs | Cat#: E7760 |
| RNA-Scope H2O2 & Protease Plus Reagents Kit | Advanced Cell Diagnostics | Cat#: 322330 |
| RNA-Scope 2.5 HD Duplex Detection Reagents Kit | Advanced Cell Diagnostics | Cat#: 322500 |
| RNA-Scope Probe Mm-GCH1-C2 | Advanced Cell Diagnostics | Cat#: 452601-C2 |
| RNA-Scope Negative Control Probe | Advanced Cell Diagnostics | Cat#: 310043 |
| Tn5 transposase | Illumina | Cat#: 20034197 |
| MinElute Reaction Cleanup Kit | QIAGEN | Cat#: 28206 |
| NEBNext High Fidelity 2 × PCR Master Mix | New England Biolabs | Cat#: M0541S |
| SPRIselect beads | Beckman Coulter | Cat#: B23319 |
| BCA Protein Assay Kit | FUJIFILM Wako | Cat#: 297-73101 |

## Patient samples

Immunohistochemistry for YAP/TAZ was performed using tissue array sections from 195 patients with lung adenocarcinoma, as previously reported (Matsubara et al, 2023). All patients provided written informed consent to participate in the study, which was approved by the Medical Ethics Committee of Kumamoto University (approval no: 1174). Patients were divided into two groups according to their YAP/TAZ expression pattern. Cases with more than 80% of cancer cells exhibiting YAP/TAZ[high] (marked accumulation and nuclear localization of YAP/TAZ) expression were classified as the "homogenous" group, while cases with less than 80% of cancer cells exhibiting YAP/TAZ[high] expression were classified as the "heterogeneous" group. The association between YAP/TAZ expression patterns and clinicopathological factors in patients with lung adenocarcinoma is summarized in Table EV1.

## Tumor transplantation model

Eight to 10-week-old female mice were used in this study. The mouse experimental procedures were approved by the Kumamoto University Animal Experiment Committee and performed in accordance with the laws and regulations concerning animal experiments, animal care and maintenance standards, and basic guidelines (approval no: A2023-031). The animals were housed in a specific pathogen-free area at the Center for Animal Resources and Development, Kumamoto University. LLC cells ($5 \times 10^5$) were subcutaneously transplanted into the back flanks of C57BL/6JJcl mice (Kyudo, Japan), and drug administration was initiated 5 days after transplantation. Tumor-bearing mice were treated with intratumoral injections of vehicle (10% DMSO, 45% PEG300, and 45% PBS), DAHP (100 mg/kg body weight in the vehicle), or RSL3 (50 mg/kg body weight in the vehicle) thrice a week for the entire duration of the experiment until the mice were euthanized. Tumor height and width were measured using a caliper every 2 days to calculate tumor volume (tumor volume width$^2$ × height × 0.523). Mice were euthanized when the tumor diameter exceeded 15 mm or when ulceration occurred. For some experiments,

tumors were harvested and weighed after treatment for two weeks, followed by histological analysis.

## Immunohistochemical analysis

Tumor tissues embedded in paraffin were sectioned into 4-μm slices. After deparaffinization, the tumor sections were rehydrated using xylene and alcohol gradients and subjected to heat-induced antigen retrieval in citrate buffer (1.8 mM citric acid monohydrate, 8.2 mM sodium citrate, pH 6.0). The sections were incubated overnight at 4 °C with anti-YAP/TAZ antibody (1:100, #93622; Cell Signaling Technology, Danvers, MA, USA), followed by incubation with a secondary antibody using the VECTASTAIN Elite ABC Rabbit IgG Kit (#PK-6101; Vector Laboratories, Newark, CA, USA). The Peroxidase Stain DAB Kit (#25985-50; Nacalai Tesque, Kyoto, Japan) was used as the chromogenic substrate, and the sections were counterstained with hematoxylin.

For representative cases with diffuse and heterogeneous YAP/TAZ expression patterns, consecutive staining on the same slides was performed using anti-cytokeratin (CK; clone AE1/AE3; #M3515; Agilent Technologies, Santa Clara, CA, USA) and anti-YAP/TAZ antibody (1:100, #93622; Cell Signaling Technology, Danvers, MA, USA). 3-amino-9-ethylcarbazole (AEC; #415182; Nichirei, Tokyo, Japan) was used to visualize the immunoreactions, and the preceding steps were conducted in the same manner as described above. After the first round of antibody staining, destaining and antibody stripping were performed as follows, based on the method previously described (Glass et al, 2009). The slides were decoverslipped in tris-buffered saline (TBS), dehydrated in an alcohol gradient to 95% ethanol, and then incubated in 95% ethanol until the AEC reaction product was no longer visible. The slides were subsequently rehydrated through 70% ethanol and transferred to the appropriate antigen retrieval buffer. They were then irradiated in a microwave at full power until the solution boiled. After cooling down and two washes in TBS, the slides underwent the next round of antibody staining, starting with the blocking step. The slides were digitized using a NanoZoomer S20 (Hamamatsu Photonics, Hamamatsu, Japan) whole-slide scanner, and subsequent digital image analysis was performed using HALO (version 4.0.5107.318; Indica Labs, Albuquerque, NM, USA). Pseudocolored images containing staining intensities for both hematoxylin and AEC were generated using the color deconvolution function. Finally, multichannel pseudocolored images were created by fusing the images for YAP/TAZ staining and CK (AE1/AE3) staining.

## Cell culture and stimulation

LLC, LK-2, PC-9, and H1975 cells were cultured in RPMI-1640 medium (#189-02025; Wako) supplemented with 10% fetal bovine serum (FBS) (#175012; Nichirei, Tokyo, Japan), penicillin (100 U/mL) (#168-23191; Wako), and streptomycin (100 μg/mL) (#168-23191; Wako) under an atmosphere of 5% $CO_2$ at 37 °C. A549 cells were cultured under the same conditions, except RPMI-1640 was replaced with DMEM (#044-29765; Wako). The cells were plated at a density of $5 \times 10^4$ cells/well in 12-well plates and incubated for 16 h unless otherwise noted. Cells were treated with the following reagents for the indicated time: (1S,3 R)-RSL3 (400 nM, #S8155; Selleck Chemicals, Houston, TX, USA), Erastin (5 μM, #17754;

Cayman Chemical), L-Buthionine-(S,R)-Sulfoximine (L-BSO, 10 mM, #14484; Cayman Chemical), Sulfasalazine (SAS, 1 mM, #HY-14655; MedChemExpress), actinomycin D (500 nM, #018-21264; Wako), TNF-α (30 ng/mL, #ab206641; Abcam, Cambridge, UK), cycloheximide (30 μg/mL, #037-20991; Wako), Z-VAD-FMK (50 μM for apoptosis inhibition, 20 μM for necroptosis induction, #S7023; Selleck Chemicals), Necrostatin-1 (50 μM, #S8037; Selleck Chemicals), Ferrostatin-1 (5 μM, #17729; Cayman Chemical, Ann Arbor, MI, USA), (6 R)-5,6,7,8-tetrahydro-L-biopterin (BH4, 40 μM, #81880; Cayman Chemical), 7,8-dihydro-L-biopterin (BH2) (40 μM, #81882; Cayman Chemical), and DAHP (10 mM, #81260; Cayman Chemical). For cysteine and cystine starvation (CCS) experiments, cells were cultured in RPMI 1640 medium without L-glutamine, L-cysteine, L-cystine, and L-methionine (#1646454; MP Biomedicals) supplemented with 10% dialyzed FBS (#S-FBS-NL-065; Serana), L-glutamine (2 mM, #073-05391; Wako), and L-methionine (0.1 mM, #13038-72; Nacalai) for 24 h. For serum starvation experiments, cells were seeded at $4 \times 10^3$ cells (for serum-repleted samples) or $1.6 \times 10^4$ cells (for serum-depleted samples) per one well of 12-well plates and incubated in 10% FBS for 16 h at 37 °C. Cells were then cultured in 0 or 10% FBS for an additional 32 h at 37 °C. The resulting cell densities of serum-repleted and serum-depleted samples were comparable. For high-cell-density experiments, cells were plated at $2.5 \times 10^3$ (low density) or $2.5 \times 10^5$ (high density) cells/well into 12-well plates and cultured for 52 h at 37 °C.

## Gene deletion by CRISPR/Cas9 system

YAP, TAZ, and GCH1 knockout cells were generated using the CRISPR/Cas9 system. Guide RNA sequences were designed using CRISPOR (http://crispor.tefor.net/) (Concordet and Haeussler, 2018). The guide sequences used were 5′-CATCAGATCGTG CACGTCCG-3′ for human *YAP (YAP1)*; 5′-TGTCTAGGTCCTG CGTGACG-3′ for human *TAZ (WWTR1)*; 5′-GGCAGCTTGC GAAGCCGCAT-3′ for mouse *Yap (Yap1)*; 5′-GGATTGGCGC GAGTGCGAGC-3′ for mouse *Taz (Wwtr1)*; and 5′- GCTCGGAG AACCCATTGGTG-3′ for mouse *Gch1*. Targeted sequences for mouse YAP and TAZ were cloned into the PX459 plasmid (#62988; Addgene). The targeting sequence for human YAP, human TAZ and mouse GCH1 was cloned into a modified lentiCRISPR v2-Puro plasmid, which was derived from the lentiCRISPR v2-Blast plasmid (#83480; Addgene, Watertown, MA, USA) by replacing the hygromycin resistance gene with a puromycin resistance gene. YAP/TAZ-targeting PX459 plasmids were transfected into LLC cells using the Lipofectamine 3000 transfection kit (#L3000-08; Invitrogen, Waltham, MA, USA) to delete YAP/TAZ in LLC cells according to the manufacturer's instructions. After transfection and transient selection with puromycin (5 μg/mL, #A1113803; Thermo Fisher Scientific, Waltham, MA, USA) for 3 days, cells were single-cell sorted using a cell sorter (SH800S, Sony, Tokyo, Japan) and cultured for expansion. Knockout clones were selected using immunoblot analysis because of the lack of target protein expression. To delete YAP/TAZ in LK-2 cells or GCH1 in LLC cells, YAP/TAZ-targeting lentiCRISPR v2-Puro plasmids or GCH1-targeting lentiCRISPR v2-Puro plasmids were co-transfected with the packaging plasmids pLenti-P2A and pLenti-P2B (#LV003; Applied Biological Materials, Vancouver, Canada) into HEK293T cells using PolyJet (#SL100688; SignaGen, Frederick,

MD, USA), following the manufacturer's instructions. Lentiviral supernatant was supplemented with 5 μg/mL polybrene (#TR-1003-G; Merck, Rahway, NJ, USA) after 48 h of transfection, filtered through a 0.45-μm filter (#IPVH00010; Merck), and used for infection. Cells were selected with puromycin (5 μg/mL) in a culture medium for 3 days after 48 h of infection.

## Lentiviral infection

Lentiviral infection was performed as previously described (Thinyakul et al, 2024). Briefly, a pLVSIN-CMV empty vector (#6183; Takara, Kusatsu, Japan), pLVSIN-CMV HA-YAP, pLVSIN-CMV EGFP, or pLVSIN-CMV tdTomato plasmids were co-transfected with packaging plasmids pLenti-P2A and pLenti-P2B into HEK293T cells using PolyJet, followed by lentiviral infection and puromycin selection, as described above. EGFP or tdTomato-positive cells were sorted using a cell sorter (FACSAriaIIIu, BD Biosciences, Franklin Lakes, NJ, USA) 48 h after infection to select fluorescent protein-expressing cells.

## Analysis of cell viability and lipid peroxidation by flow cytometry

Flow cytometry was performed using FACSVerse (BD Biosciences) or CytoFLEX S (Beckman Coulter, Brea, CA, USA), and the results were analyzed using FlowJo software (BD Biosciences). The cells were suspended in FACS buffer (2% FBS in phosphate-buffered saline [PBS]) and stained with 0.1 μg/mL propidium iodide (PI; #341-07881; Dojindo, Kumamoto, Japan) to evaluate cell viability. The percentage of dead PI-positive cells was measured using flow cytometry. Cells were suspended in Hanks' Balanced Salt Solution (HBSS; #084-08345; FUJIFILM Wako) containing 5 μM BODIPY 581/591 C11 (#D3861; Thermo Fisher Scientific) and incubated for 15 min at 37 °C to assess lipid peroxidation. The cells were then washed once with HBSS and suspended in FACS buffer to evaluate lipid peroxide levels using a flow cytometer. The percentage of cells positive for fluorescent signals was calculated.

## Coculture experiment of fluorescent protein-expressing cells

EGFP- or tdTomato-expressing LLC cells were cultured and stimulated with 400 nM RSL3 for 10 h. Fluorescent images were obtained using BZ-X800 (KEYENCE, Osaka, Japan), and the number of cells expressing fluorescent proteins were counted at 0 and 10 h after RSL3 treatment. The cell number was normalized to the average cell number of five randomly selected images per plate and multiplied by 100% to obtain the relative cell number values. BZ analyzer (KEYENCE) software was used for quantitative analysis.

## Reverse transcription (RT) and real-time PCR analysis

Total RNA was extracted from cells using TRIzol reagent (#15596026; Invitrogen) and reverse-transcribed to complementary DNA using the ReverTra Ace qPCR RT kit (#FSQ-301; TOYOBO, Osaka, Japan). Complementary DNA was diluted and subjected to real-time PCR using a StepOnePlus Real-Time PCR System

(Thermo Fisher Scientific). The SYBR qPCR Mix (#QPX-201; TOYOBO) was used for PCR according to the manufacturer's protocol. PCR primer sequences (forward and reverse, respectively) were 5′-CGATGAGTATAAACGGACGGCACTC-3′ and 5′-GGATGGCTGTGGATTCAAGCATATC-3′ for mouse *Ankrd1*; 5′-AGCAAGTCCTTGGTCTCAGTAAAC-3′ and 5′-ACCGCAATCTGTTTGGTGAGGC-3′ for mouse *Gch1*; 5′-GAAGTCCAGTGTGGGAACAGGT-3′ and 5′-CAACCACTCAGTGGCACCAACA-3′ for mouse *Tfrc*; 5′-TCCCTGTCCTACGGCAATGAAG-3′ and 5′-CTGGAACACGAAGACCACAAGG-3′ for mouse *Emp1*; 5′-CCTTTGGCTCATGTGCTGGAAC-3′ and 5′-GCCATAAGTGTGGGTTTCAGTAC-3′ for mouse *Acsl4*; 5′-CTGGACAGTGATTCAGAGACGC-3′ and 5′-GATGCTGTGCATCTTTTCCAGGC-3′ for mouse *Angptl4*; 5′-GCCTGGAGAAACCTGCCAAGTATG-3′ and 5′-GAGTGGGAGTTGCTGTTGAAGTCG-3′ for mouse *Gapdh*. Reactions for *Gapdh* mRNA were performed concurrently on the same plate as those for the test mRNAs, and the results were normalized to the corresponding amount of *Gapdh* mRNA.

## Immunoblotting

Equal amounts of protein samples were resolved using sodium dodecyl-sulfate polyacrylamide gel electrophoresis under reducing conditions. After transferring the samples to polyvinylidene fluoride membranes (#IPVH00010; Millipore, Burlington, MA, USA), the following antibodies were used to detect proteins: anti-YAP/TAZ (1:2000, #sc-101199; Santa Cruz, Santa Cruz, CA, USA), anti-YAP (1:2000, #14074; Cell Signaling Technology), anti-GCH1 (1:2000, #PA5-120098; Thermo Fisher Scientific), anti-HA-Tag (1:2000, #2999; Cell Signaling Technology), anti-β-actin (1:2000, #12620; Cell Signaling Technology), horseradish peroxidase (HRP)-linked mouse IgG (1:5000, #7076; Cell Signaling Technology), or HRP-linked rabbit IgG (1:5000, #7074; Cell Signaling Technology). Immunoblot signals were detected using a chemiluminescent HRP substrate (#WBKLS0500; Millipore), and images were obtained using the ChemiDoc Touch imaging system (#17001401JA; Bio-Rad Laboratories). Phos-tag electrophoresis was performed as previously described (Moroishi et al, 2015b). The chemiluminescence intensity was quantified by scanning using ImageJ software (NIH).

## RNA-sequencing analysis

Total RNA was extracted from LLC WT or YAP/TAZ dKO cells using the RNeasy Mini Kit (#74104; QIAGEN, Hilden, Germany). Three biological replicates were analyzed for each sample. Libraries were constructed from total RNA using the NEBNext Poly(A) mRNA Magnetic Isolation Module (#E7490; New England Biolabs, Ipswich, MA, USA) and NEBNext Ultra II Directional RNA Library Prep Kit (#E7760; New England Biolabs). High-throughput sequencing was performed using a NovaSeq 6000 system (Illumina, San Diego, CA, USA), and 150-base pair paired-end reads were generated. Fastp (v. 0.23.2) (Chen, 2023) was used to ensure the high per-base-sequence quality of the reads. Paired-end reads were aligned to the mouse (mm10) genome using the STAR software (v. 2.7.10b) (Dobin et al, 2013). DESeq2 (version 1.42.1) (Love et al, 2014) was used to identify differentially expressed genes. A volcano plot was constructed using EnhancedVolcano (v. 1.20.0) (Blighe et al, 2024). Ferroptosis-related genes were selected using WikiPathway (Agrawal et al, 2024) (Ferroptosis, WP4313) and are listed in Table EV2.

## RNAscope

LLC subcutaneous tumors were harvested 14 days after transplantation. Tumor tissues were fixed in 10% neutral buffered formalin for 48 h at 4 °C, followed by embedment in paraffin. The tumor tissues were sectioned into 4 μm slices and air-dried overnight at 42 °C. RNA-Scope in situ hybridization was performed using the RNA-Scope H2O2 & Protease Plus Reagents Kit (#322330; Advanced Cell Diagnostics, Minneapolis, MN, USA) and RNA-Scope 2.5 HD Duplex Detection Reagents Kit (#322500; Advanced Cell Diagnostics) following the manufacturer's instructions. *Gch1* was detected using the RNA-Scope Probe Mm-GCH1-C2 (#452601-C2; Advanced Cell Diagnostics), and the RNA-Scope Negative Control Probe (#310043; Advanced Cell Diagnostics) was used as a negative control. Images from five randomly selected fields were obtained and quantified using the BZ analyzer software (KEYENCE).

## Assay for transposase-accessible chromatin with sequencing

Cells were isolated and subjected to assay for transposase-accessible chromatin with high-throughput sequencing (ATAC-seq) as previously described (Kurotaki et al, 2019). Briefly, LLC WT or YAP/TAZ dKO cells ($1 \times 10^4$) were lysed using 20 μL of cold lysis buffer [10 mM Tris-HCl (pH 7.4), 10 mM NaCl, 3 mM MgCl$_2$, 0.1% IGEPAL CA-630, 0.1% Tween20, and 0.01% digitonin] to prepare nuclei. The nuclei were then incubated with 20 μL of transposase solution [10 μL of $2 \times$ Tagment DNA buffer: 20 mM Tris-HCl (pH 7.6), 10 mM MgCl$_2$, 20% Dimethylformaldehyde; 6.6 μL of PBS, 0.1% Tween20, 0.01% Digitonin, 2 μL of Tn5 transposase (Illumina, #20034197)] at 37 °C for 30 min with shaking at 900 rpm. DNA was purified using a MinElute Reaction Cleanup Kit (#28206; QIAGEN). The purified DNA was amplified using NEBNext High Fidelity 2× PCR Master Mix (#M0541S; New England Biolabs) with indexed primers. The amplified DNA was further purified using SPRIselect beads (#B23319; Beckman Coulter) to remove primer dimers. The prepared libraries were sequenced on the NextSeq500 platform (Illumina) to generate paired-end 38 bp reads. ATAC-seq data were obtained from three biological replicates. Bowtie software (v. 2.5.0) was used to map the ATAC-seq reads to the mouse reference genome sequence (mm10/GRCm38). The mapped SAM format files were converted into tag directories using the makeTagDirectory module of the Homer package, and duplicate reads were removed using the '-tbp 1' option. The findPeaks module of the Homer package was used to identify the ATAC-seq peaks. The HOMER command options used in this study were as follows: cmd = findPeaks [tagdirectory] -center -size 150 [output peak file].

## Measurement of BH4 and BH2

The BH4 and BH2 levels were measured using liquid chromatography-electrospray ionization-mass spectrometry (LC-MS) with an Agilent 6460 Triple Quadrupole LC-MS system (Agilent Technologies, Santa Clara, CA, USA). For measurement of intracellular BH4 and BH2, LLC cells ($7.25 \times 10^5$) were cultured in 100 mm tissue culture dishes for 16 h, washed once with wash buffer (50 mM ascorbic acid, 6.5 mM DTT, and 1 mM EDTA in

PBS), and harvested in borate buffer (50 mM ascorbic acid, 10 mM DTT, and 1 mM EDTA, pH 8.9). The cell suspension was homogenized at 250 W in 10-s intervals for 3 min using a Bioruptor UCD-250 (Tosho Electronic) and then centrifuged at $15,500 \times g$ for 10 min at 4 °C. The supernatants were diluted in water and subjected to LC-MS/MS analysis as described previously (Zhang et al, 2021). Precipitates were resuspended in 1% sodium dodecyl-sulfate in PBS, homogenized at 250 W at 10 s intervals for 10 min with a Bioruptor, and used for protein quantification with the BCA Protein Assay Kit (#297-73101; FUJIFILM Wako). For measurement of extracellular BH4 and BH2, LLC cells ($2 \times 10^6$) were cultured in 100 mm tissue culture dishes with 10 mL medium for 16 h. The culture medium was subjected to centrifugation with Amicon Ultra 3 K device (Merck Millipore) for 30 min at room temperature. The flowthrough was 3-times diluted by 0.1% FA (pH 6.0) containing 50 mM ascorbic acid, and then were used for LC-MS/MS analyses.

LC-MS/MS conditions were as follows: column, YMC-Triart C18 Plus column ($2.1 \times 50$ mm) (YMC., Kyoto, Japan); column temperature, 45 °C; injection volume, 1 μL; mobile phases: A, 0.1% formic acid (pH 6.0, adjusted by ammonium hydrogen carbonate), and B, 40% acetonitrile; gradient (B concentration), 0 min—1%, 10 min—80%, 10.1 min—1%, 25 min—1%; and flow rate, 0.2 mL/min. The general conditions for electrospray ionization mass spectrometry were nebulizer gas, nitrogen, delivered at 50 psi; nebulizer gas temperature, 250 °C; capillary voltage, 3500 V; collision gas, and G1 grade, nitrogen (Taiyo Nippon Sanso Corporation, Tokyo, Japan). Table EV3 provides details of the multiple reaction monitoring parameters used in the study.

## Glutathione measurement

Intracellular glutathione (GSH) was measured by using liquid chromatography-electrospray ionization-mass spectrometry with the Agilent 6460 Triple Quadrupole LC-MS/MS system (Agilent Technologies, Santa Clara, CA, USA). As reported previously (Zhang et al, 2019), we employed β-(4-hydroxyphenyl)ethyl iodoacetamide (HPE-IAM) as a GSH trapping agent. In brief, LLC WT and YAP/TAZ dKO cells were pre-seeded in 12-well plate as $5 \times 10^4$ cells/well. After cultured 16 h, the cells were washed with PBS once and harvested in methanol containing 5 mM HPE-IAM. The samples were then homogenized by using Bioruptor UCD-250 (Tosho Electronic, Tokyo, Japan) for 2 min, followed by an incubation at 37 °C. Thirty minutes later, supernatants were separated via centrifugation and were diluted with 0.1% formic acid containing known amounts of isotope-labeled standards. Mixtures were then subjected to LC-MS/MS for analyses. Precipitates were resuspended with 1% SDS in PBS and were then homogenized with a Bioruptor for 10 min, after which samples were used for quantifying protein concentrations with the BCA Protein Assay Kit (FUJIFILM Wako Pure Chemical Corporation). LC-MS/MS conditions were as follows: column, YMC-Triart C18 Plus column ($2.1 \times 50$ mm) (YMC Co. Ltd., Kyoto, Japan); column temperature, 45 °C; injection volume, 10 μL; mobile phases: A, 0.1% formic acid, and B, acetonitrile; gradient (B concentration), 0 min–1%, 10 min–80%, 10.5 min–1%, 15 min–1%; and f low rate, 0.2 mL/min. The general conditions for ESI-MS were nebulizer gas, nitrogen, delivered at 50 psi; nebulizer gas temperature, 250 °C; capillary voltage, 3500 V; collision gas, and G1 grade, nitrogen

(Taiyo Nippon Sanso Corporation, Tokyo, Japan). Table EV3 provides details of the multiple reaction monitoring parameters used in the study.

## Statistics and reproducibility

The correlation analysis of candidate gene expression was performed using the GEPIA (Gene Expression Profiling Interactive Analysis) tool (http://gepia.cancer-pku.cn/), which is based on datasets from The Cancer Genome Atlas (TCGA) (Tang et al, 2019). The correlation between GCH1 expression and the YAP/TAZ transcriptional target signature was evaluated across TCGA normal datasets. Statistical analyses were performed using Graph-Pad Prism 10 software (GraphPad Software, La Jolla, CA, USA). The statistical parameters and methods are reported in the figures and figure legends. Differences were considered statistically significant at $P < 0.05$. No statistical methods were used to predetermine the sample sizes; however, the sample sizes in this study were similar to those generally used in the field. The investigators were not blinded to the allocation during either the experiment or outcome assessment.

## Data availability

All RNA-seq and ATAC-seq raw data generated in this study have been deposited in the DNA Data Bank of Japan (DDBJ; accession number DRA018261).

The source data of this paper are collected in the following database record: biostudies:S-SCDT-10_1038-S44319-025-00515-4.

## Peer review information

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

## Acknowledgements

We thank Saisai Liu, Yudai Ohta, Takashi Matsumoto, Ayaka Yoshida, Eerdunduleng, Tomoko Matsunaga, and Kozue Terada for technical support in the experiments; Yanliang Liu for technical support in the RNA-seq analysis; and Center for Animal Resources and Development (CARD), Kumamoto University, for support with animal experiments. This study was supported by Japan Agency for Medical Research and Development (AMED) PRIME (grant JP22gm6210030 to TM), Japan Society for the Promotion of Science (JSPS) KAKENHI grants (22H05635, 23K18098, 24H00864, and 24H00865 to TM), Japan Science and Technology Agency (JST) FOREST Program (JPMJFR226J to TM), Kobayashi Foundation for Cancer Research (to TM), Princess Takamatsu Cancer Research Fund (to TM), Ichiro Kanehara Foundation (to TM), Foundation for Promotion of Cancer Research in Japan (to TM), and Kato Memorial Bioscience Foundation (to TM). Additional support, including financial and technical assistance, was provided by Medical Research Center Initiative for High Depth Omics (Science Tokyo), Nanken-Kyoten (Science Tokyo), and Multilayered Stress Diseases (JPMXP1323015483; Science Tokyo).

## Author contributions

**Hao Li**: Conceptualization; Data curation; Formal analysis; Investigation; Visualization; Methodology; Writing—original draft; Writing—review and editing. **Yohei Kanamori**: Data curation; Formal analysis; Investigation; Visualization; Methodology; Writing—original draft; Writing—review and editing. **Akihiro Nita**: Data curation; Formal analysis; Methodology; Writing—review and editing. **Ayato Maeda**: Formal analysis; Writing—review and editing. **Tianli Zhang**: Investigation; Methodology; Writing—review and editing. **Kenta Kikuchi**: Investigation; Methodology; Writing—review and editing. **Hiroyuki Yamada**: Formal analysis; Investigation; Writing—review and editing.

**Touya Toyomoto**: Investigation; Writing—review and editing. **Mohamed Fathi Saleh**: Formal analysis; Writing—review and editing. **Mayumi Niimura**: Investigation; Writing—review and editing. **Hironori Hinokuma**: Investigation; Writing—review and editing. **Mayuko Shimoda**: Investigation; Writing—review and editing. **Koei Ikeda**: Investigation; Writing—review and editing. **Makoto Suzuki**: Supervision; Writing—review and editing. **Yoshihiro Komohara**: Formal analysis; Supervision; Writing—review and editing. **Daisuke Kurotaki**: Formal analysis; Supervision; Methodology; Writing—review and editing. **Tomohiro Sawa**: Supervision; Methodology; Writing—review and editing. **Toshiro Moroishi**: Conceptualization; Resources; Formal analysis; Supervision; Funding acquisition; Visualization; Methodology; Writing—original draft; Project administration; Writing—review and editing.

Source data underlying figure panels in this paper may have individual authorship assigned. Where available, figure panel/source data authorship is listed in the following database record: biostudies:S-SCDT-10_1038-S44319-025-00515-4.

## Disclosure and competing interests statement

The authors declare no competing interests.

# Expanded View Figures

**Figure EV1. Generating yes-associated protein (YAP)/transcriptional co-activator with PDZ-binding motif (TAZ) (YAP/TAZ)-deficient Lewis lung carcinoma (LLC) cells.**

(A) DNA mutation in YAP/TAZ double-knockout (dKO) LLC cells using the clustered regularly interspaced short palindromic repeat (CRISPR)/Cas9 system. gDNA, genomic DNA. (B) Immunoblotting (IB) of cell extracts from wild-type (WT) and YAP/TAZ dKO LLC cells with antibodies against the indicated proteins. (C) Reverse transcription (RT) and real-time PCR analysis of the YAP/TAZ target gene *Ankrd1* in WT and YAP/TAZ dKO LLC cells. Data are means ± SD of three biologically independent samples from a representative experiment. ****$P = 0.000001244429960$ (unpaired *t* test).

▶

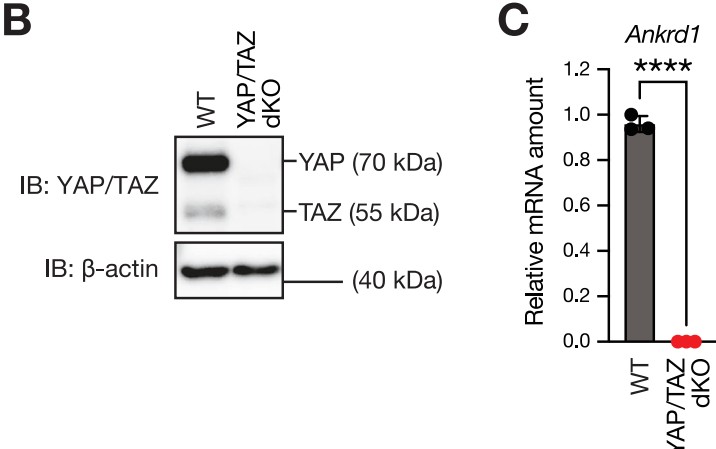

**A**

YAP gDNA

T T G A A G A A G G A G T C G G G C A G C T T G C G A A G C C G C A T GGG C A C G G T C T G A G G C A C G T T G G

Guide sequence PAM

YAP/TAZ dKO

T T G A A G A A G G A G T C G G G C A G C T T G C G A A G C C → DNA mutation

TAZ gDNA

T G A T G A G T C T G T G C T G G A T T G G C G C G A G T G C G A G C CGG A A T C G G G C T C C T T A A A G A A G

Guide sequence PAM

YAP/TAZ dKO

T G A T G A G T C T G T G C T G G A T T G G C G C G A G T G → DNA mutation

**B**

IB: YAP/TAZ

WT YAP/TAZ dKO

YAP (70 kDa)
TAZ (55 kDa)

IB: β-actin (40 kDa)

**C**

*Ankrd1*

Relative mRNA amount

****

WT YAP/TAZ dKO

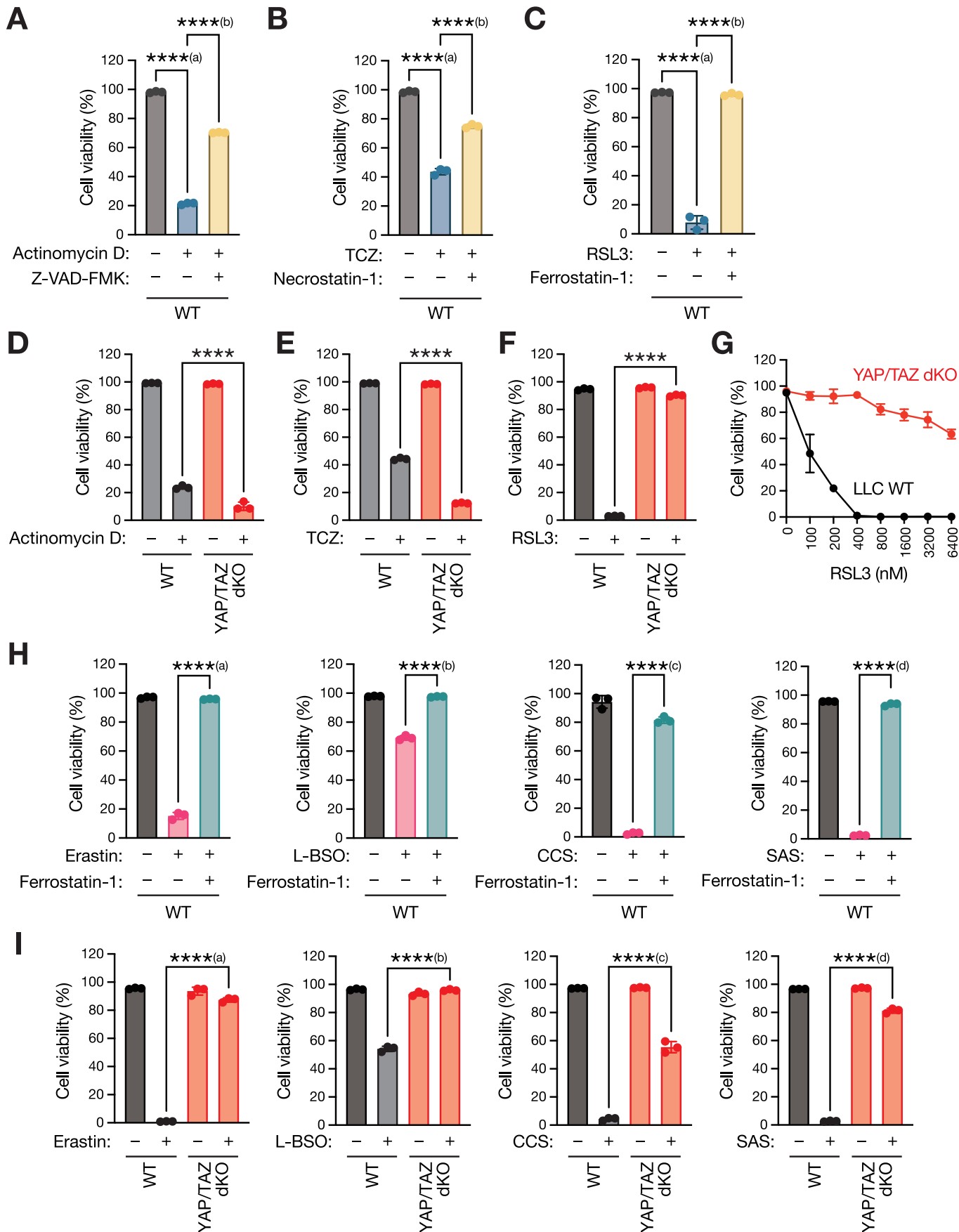

◀ **Figure EV2. YAP/TAZ-deficient LLC cells were resistant to ferroptosis.**

(A) Actinomycin D-induced cell death was rescued by the apoptosis inhibitor Z-VAD-FMK. Wild-type (WT) LLC cells were pretreated (or not) with Z-VAD-FMK (50 μM) for 1 h, followed by stimulation with Actinomycin D (500 nM) for 24 h. Dead cells were stained with propidium iodide (PI), and the percentage of the PI-negative live cell population was calculated using the flow cytometer. Data are means ± SD of three biologically independent samples from a representative experiment. ****[a]$P = 0.000000000000065$; ****[b]$P = 0.000000000000065$ (one-way ANOVA test followed by Tukey's multiple comparison test). (B) TCZ-induced cell death was rescued by the necroptosis inhibitor Necrostatin-1. WT LLC cells were pretreated (or not) with Necrostatin-1 for 1 h, followed by stimulation with TCZ [T, TNFα (30 ng/mL); C, cycloheximide (30 μg/mL); Z, Z-VAD-FMK (20 μM)] for 24 h. The percentage of live cell population was calculated using PI and flow cytometer. Data are means ± SD of three biologically independent samples from a representative experiment. ****[a]$P = 0.000000012723845$; ****[b]$P = 0.000000720360306$ (one-way ANOVA test followed by Tukey's multiple comparison test). (C) RSL3-induced cell death was rescued by the ferroptosis inhibitor Ferrostatin-1. WT LLC cells were pretreated (or not) with Ferrostatin-1 (5 μM) for 30 min, followed by stimulation with RSL3 (400 nM) for 10 h. The percentage of live cell population was calculated using PI and flow cytometer. Data are means ± SD of three biologically independent samples from a representative experiment. ****[a]$P = 0.000000048729652$; ****[b]$P = 0.000000056548852$ (one-way ANOVA test followed by Tukey's multiple comparison test). (D) YAP/TAZ loss sensitizes LLC cells to apoptosis. WT and YAP/TAZ dKO LLC cells were treated with Actinomycin D (500 nM) for 24 h, followed by cell viability assay using flow cytometer. Data are means ± SD of three biologically independent samples from a representative experiment. ****$P = 0.000030539048461$ (one-way ANOVA test followed by Tukey's multiple comparison test). (E) YAP/TAZ loss sensitizes LLC cells to necroptosis. WT and YAP/TAZ dKO LLC cells were treated with TCZ [T, TNFα (30 ng/mL); C, cycloheximide (30 μg/mL); Z, Z-VAD-FMK (20 μM)] for 24 h, followed by cell viability assay using flow cytometer. Data are means ± SD of three biologically independent samples from a representative experiment. ****$P = 0.000000000000054$ (one-way ANOVA test followed by Tukey's multiple comparison test). (F) YAP/TAZ loss confers resistance to ferroptosis in LLC cells. WT and YAP/TAZ dKO LLC cells were treated with RSL3 (400 nM) for 10 h, followed by cell viability assay using flow cytometer. Data are means ± SD of three biologically independent samples from a representative experiment. ****$P = 0.000000000000051$ (one-way ANOVA test followed by Tukey's multiple comparison test). (G) Cell viability analysis was performed on WT and YAP/TAZ dKO LLC cells treated with indicated concentrations of RSL3 for 10 h, followed by cell viability assay using flow cytometer. Data are means ± SD of three biologically independent samples from a representative experiment. (H) Cell death induced by ferroptosis-inducing reagents or cysteine and cystine starvation (CCS) was rescued by the ferroptosis inhibitor Ferrostatin-1. WT LLC cells were pretreated (or not) with Ferrostatin-1 (5 μM) for 30 min, followed by stimulation with Erastin (5 μM) for 10 h, L-Buthionine-(S,R)-Sulfoximine (L-BSO; 10 mM) for 24 h, or Sulfasalazine (SAS; 1 mM) for 24 h. For CCS, cells were cultured in the cysteine and cystine-starved medium for 24 h. The percentage of live cell population was calculated using flow cytometer. Data are means ± SD of three biologically independent samples from a representative experiment. ****[a]$P = 0.000000000005959$; ****[b]$P = 0.000000096903963$; ****[c]$P = 0.000000191080058$; ****[d]$P = 0.000000000000065$ (one-way ANOVA test followed by Tukey's multiple comparison test). (I) YAP/TAZ loss confers resistance to ferroptosis in LLC cells. WT and YAP/TAZ dKO LLC cells were treated with Erastin (5 μM) for 10 h, L-BSO (10 mM) for 24 h, or SAS (1 mM) for 24 h. For CCS, cells were cultured in the cysteine and cystine-starved medium for 24 h. The percentage of live cell population was calculated using flow cytometer. Data are means ± SD of three biologically independent samples from a representative experiment. ****[a]$P = 0.000000000000769$; ****[b]$P = 0.000000000883037$; ****[c]$P = 0.000000000024749$; ****[d]$P = 0.000000000000051$ (one-way ANOVA test followed by Tukey's multiple comparison test).

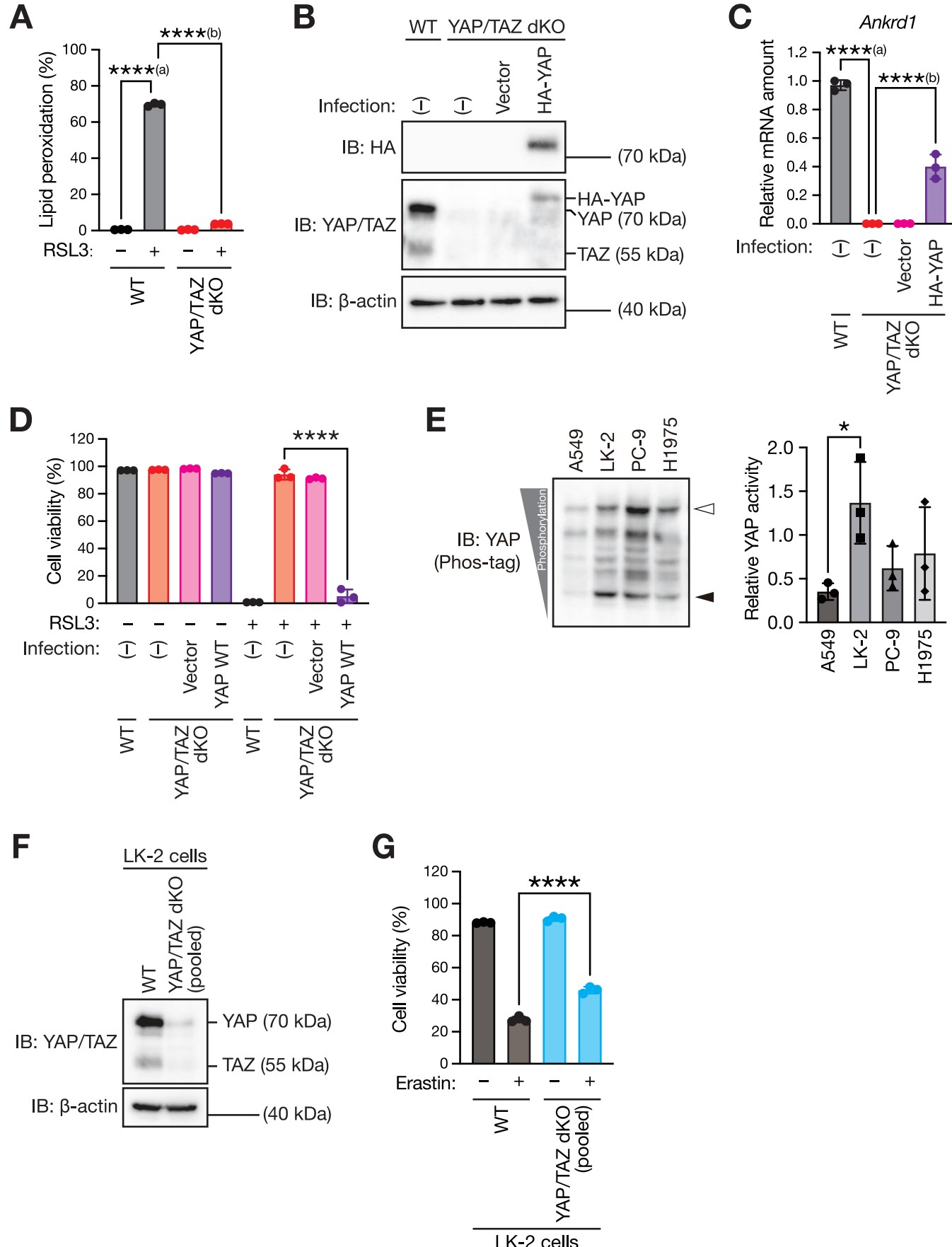

◀ **Figure EV3. YAP/TAZ determines the sensitivity to ferroptosis.**

(A) YAP/TAZ depletion suppresses lipid peroxidation in LLC cells. WT and YAP/TAZ dKO LLC cells were treated with RSL3 (400 nM) for 4 h, followed by lipid peroxidation measurement using flow cytometer. Data are means ± SD of three biologically independent samples from a representative experiment. ****[(a)]P = 0.000000000000051; ****[(b)]P = 0.000000000000051 (one-way ANOVA test followed by Tukey's multiple comparison test). (B) Immunoblot (IB) analysis of cell extracts from WT and YAP/TAZ dKO LLC cells infected (or not infected) with HA-YAP expression vectors. (C) RT and real-time PCR analysis of the YAP/TAZ target *Ankrd1* gene in WT and YAP/TAZ dKO LLC cells infected (or not infected) with HA-YAP expression vectors. Data are means ± SD of three biologically independent samples from a representative experiment. ****[(a)]P = 0.000000023517185; ****[(b)]P = 0.000027011157926 (one-way ANOVA test followed by Tukey's multiple comparison test). (D) Re-expression of YAP sensitizes YAP/TAZ-deficient LLC cells to ferroptosis. WT and YAP/TAZ dKO LLC cells infected (or not) with expression vectors for HA-YAP were treated with RSL3 (400 nM) for 10 h, followed by cell viability assay using flow cytometer. Data are means ± SD of three biologically independent samples from a representative experiment. ****P = 0.000000004002113 (one-way ANOVA test followed by Tukey's multiple comparison test). (E) Left: Representative image of Phos-tag immunoblotting (IB) analysis of cell extracts from A549, LK-2, PC-9 and H1975 human lung cancer cells using antibodies against YAP. Right: The relative activity of YAP in each cell line was assessed based on the ratio of non-phosphorylated YAP (black arrowhead) to fully phosphorylated YAP (white arrowhead). Data are means ± SD of three biologically independent samples from a representative experiment. *P = 0.044457726502518 (one-way ANOVA test followed by Tukey's multiple comparison test). (F) Immunoblotting (IB) of cell extracts from WT and YAP/TAZ dKO LK-2 cells with antibodies against the indicated proteins. (G) WT and YAP/TAZ dKO LK-2 cells were treated with Erastin (5 μM) for 10 h, followed by cell viability assay using flow cytometer. Data are means ± SD of three biologically independent samples from a representative experiment. ****P = 0.000002538628468 (one-way ANOVA test followed by Tukey's multiple comparison test).

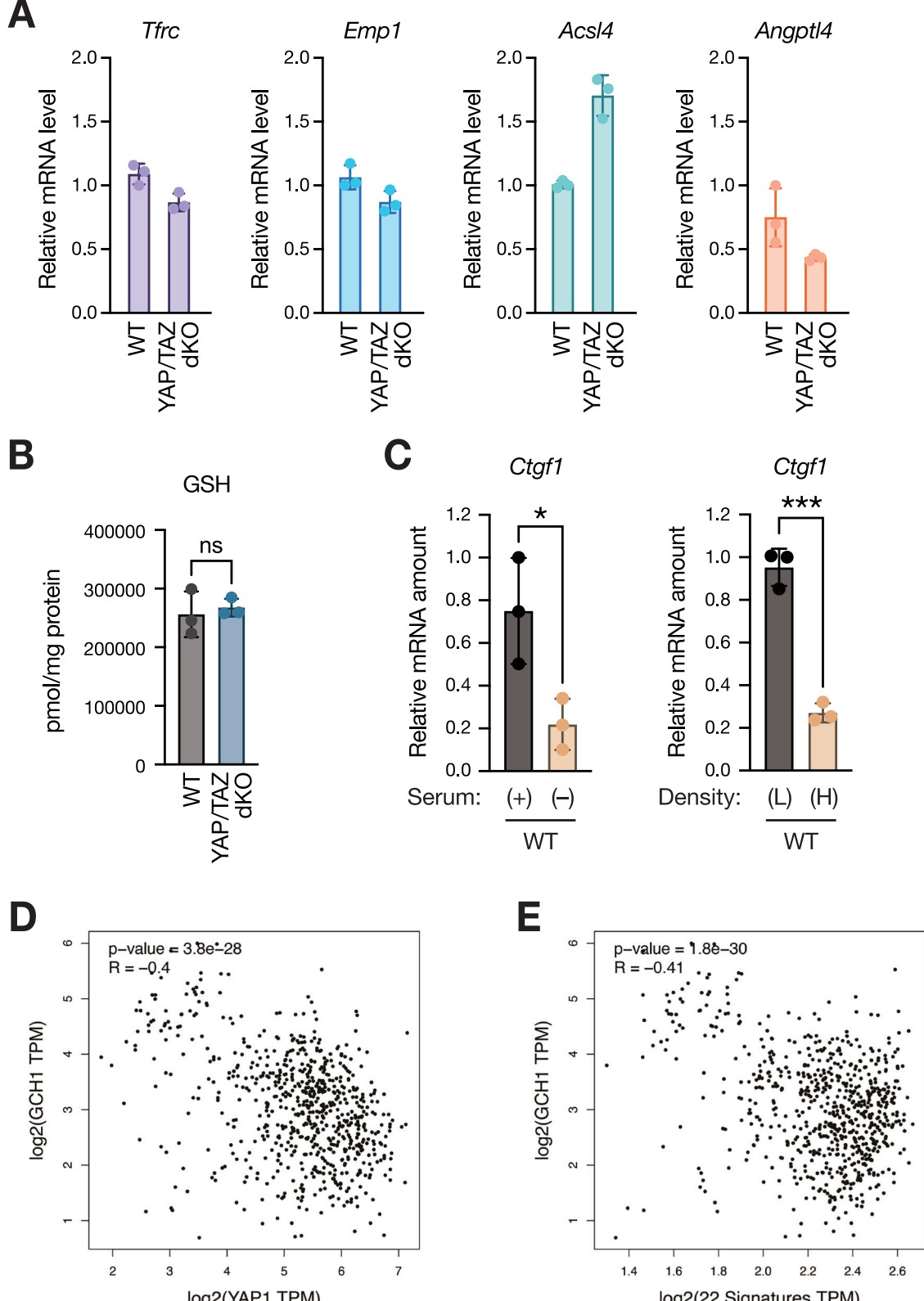

◀ **Figure EV4. Impacts of YAP/TAZ inhibition in LLC cells.**

(A) RT and real-time PCR analysis of the ferroptosis-related genes in WT and YAP/TAZ dKO LLC cells. Data are means ± SD of three biologically independent samples from a representative experiment. (B) Intracellular glutathione (GSH) levels were measured using LC-MS/MS. Data are means ± SD of three biologically independent samples from a representative experiment. ns indicates no significant difference, ns, not significant ($P = 0.658668395593572$) (unpaired $t$ test). (C) RT and real-time PCR analysis of the YAP/TAZ target *Ctgf* gene in WT LLC cells cultured with or without serum for 32 h (left) or at low (L) or high (H) cell densities for 52 h (right). Data are means ± SD of three biologically independent samples from a representative experiment. *$P = 0.029089468282210$; ***$P = 0.000272320651963$ (unpaired $t$ test). (D) The correlation between *GCH1* and *YAP1* mRNA levels across TCGA normal datasets was analyzed using the GEPIA (Gene Expression Profiling Interactive Analysis) tool with Pearson correlation. (E) The correlation between *GCH1* mRNA levels and the YAP/TAZ transcriptional target signature across TCGA normal datasets was analyzed using the GEPIA tool with Pearson correlation.

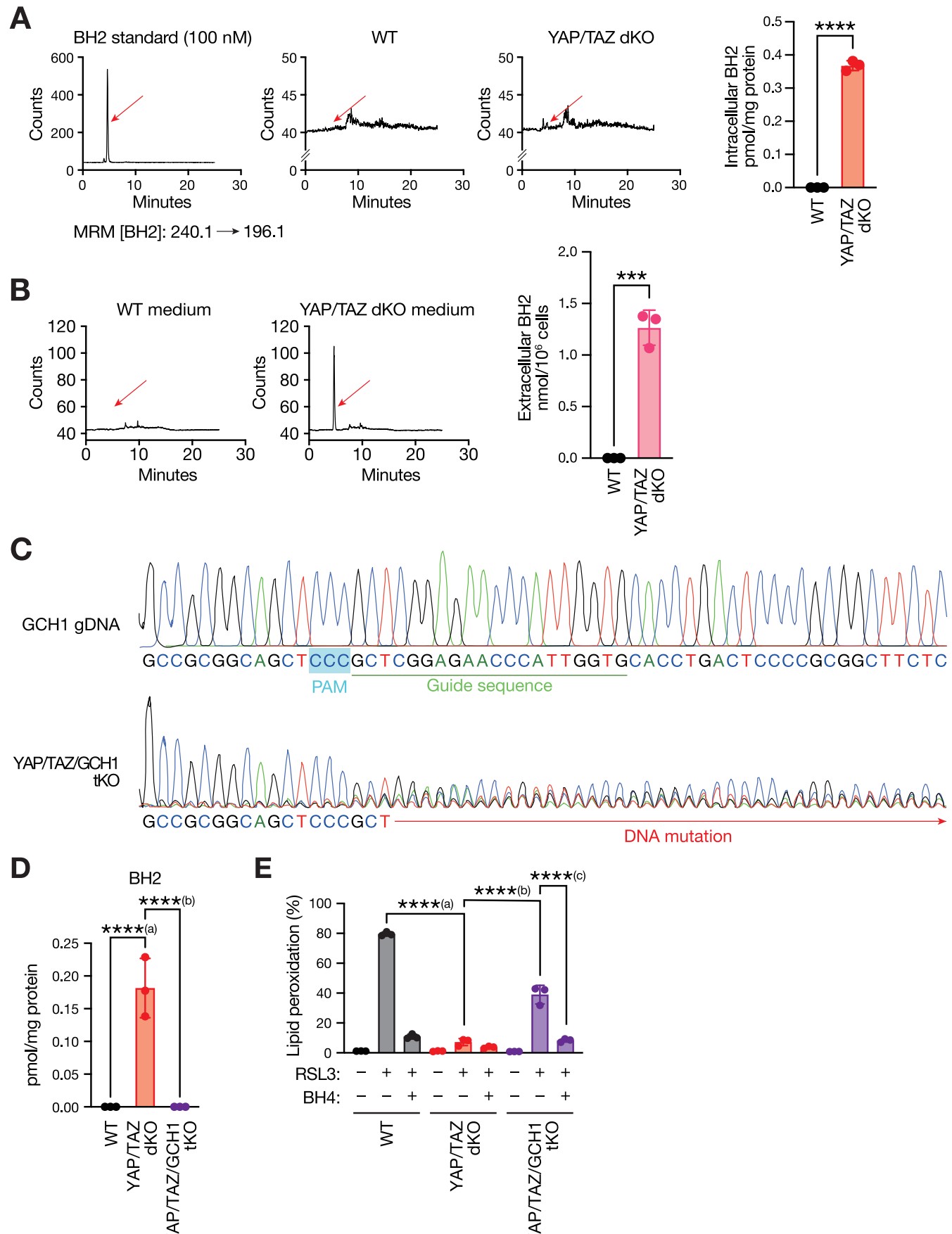

Figure EV5.  GCH1 is required for ferroptosis resistance in YAP/TAZ-deficient LLC cells.

(A) Intracellular dihydrobiopterin (BH2) levels in WT and YAP/TAZ dKO LLC cells. Peaks with the corresponding multiple reaction monitoring (MRM) indicate intracellular BH2 (left, red arrows). Quantitative analysis of BH2 levels is shown on the right. Data are means ± SD of three biologically independent samples from a representative experiment. ****$P = 0.000001790642720$ (unpaired $t$ test). (B) BH2 levels in the culture media from WT and YAP/TAZ dKO LLC cells. Peaks with the corresponding MRM indicate extracellular BH2 (red arrows). Quantitative analysis of extracellular BH2 levels is shown on the right. Data are means ± SD of three biologically independent samples from a representative experiment. ***$P = 0.000211344236570$ (unpaired $t$ test). (C) DNA mutation in YAP/TAZ/GCH1 tKO LLC cells using the CRISPR/Cas9 system. gDNA, genomic DNA. (D) BH2 levels in WT, YAP/TAZ dKO, and YAP/TAZ/GCH1 tKO LLC cells. Data are means ± SD of three biologically independent samples from a representative experiment. ****$^{(a)}P = 0.000357087917378$; ****$^{(b)}P = 0.000357087917378$ (one-way ANOVA test followed by Tukey's multiple comparison test). (E) WT, YAP/TAZ dKO, and YAP/TAZ/GCH1 triple-knockout (tKO) LLC cells were pretreated (or not) with BH4 (40 μM) for 30 min, followed by stimulation with RSL3 (400 nM) for 4 h. The percentage of cells positive for lipid peroxidation was calculated using flow cytometer. Data are means ± SD of three biologically independent samples from a representative experiment. ****$^{(a)}P = 0.000000000005554$; ****$^{(b)}P = 0.000000155138171$; ****$^{(c)}P = 0.000000230915225$ (one-way ANOVA test followed by Tukey's multiple comparison test).

