## [Peer Review File · EMBO Reports]

Hippo pathway controls biopterin metabolism to shield adjacent cells from ferroptosis in lung cancer

Hao Li, Yohei Kanamori, Akihiro Nita, Ayato Maeda, Tianli Zhang, Kenta Kikuchi, Hiroyuki Yamada, Touya Toyomoto, Mohamed Saleh, Mayumi Niimura, Hironori Hinokuma, Mayuko Shimoda, Koei Ikeda, Makoto Suzuki, Yoshihiro Komohara, Daisuke Kurotaki, Tomohiro Sawa, and Toshiro MOROISHI

Corresponding author(s): Toshiro MOROISHI (moroishi.toshiro@tmd.ac.jp)

Review Timeline:

Submission Date:	22nd Aug 24
Editorial Decision:	2nd Oct 24
Revision Received:	7th Mar 25
Editorial Decision:	6th May 25
Revision Received:	4th Jun 25
Accepted:	11th Jun 25

Transaction Report:

Dear Dr. MOROISHI

Thank you for the submission of your research manuscript to our journal. We have now received the full set of referee reports that is copied below.

As you will see, the referees acknowledge that the findings are potentially interesting, but they also raise a number of concerns that need to be addressed to strengthen your observations and conclusions. It will be important to support your observations by screening and testing additional cell lines and ferroptosis inducers. Some more data on the YAP/TAZ status and heterogeneity in patient samples, if possible, or by screening publicly available datasets should be provided and the pathophysiological relevance of your findings should be discussed and toned down as appropriate. Extracellular BH4 levels and potentially uptake mechanisms should be analysed, although the latter is not mandatory. Points 5 and 6 from Referee 2 can be addressed in the discussion. Please note that our editorial policies do not permit to refer to 'data not shown' (point 3, Referee 3). Therefore, please include these data to support your statements.

Given the constructive comments, we would like to invite you to revise your manuscript with the understanding that the referee concerns (as detailed above and in their reports) must be fully addressed and their suggestions taken on board. Please address all referee concerns in a complete point-by-point response. Acceptance of the manuscript will depend on a positive outcome of a second round of review. It is EMBO Reports policy to allow a single round of revision only and acceptance or rejection of the manuscript will therefore depend on the completeness of your responses included in the next, final version of the manuscript.

We realize that it is difficult to revise to a specific deadline. In the interest of protecting the conceptual advance provided by the work, we recommend a revision within 3 months (January 3rd). Please discuss the revision progress ahead of this time with the editor if you require more time to complete the revisions.

I am also happy to discuss the revision further via e-mail or a video call, if you wish.

*******IMPORTANT NOTE:**

We perform an initial quality control of all revised manuscripts before re-review. Your manuscript will FAIL this control and the handling will be delayed IN CASE the following APPLIES:

- 1) A data availability section providing access to data deposited in public databases is missing. If you have not deposited any data, please add a sentence to the data availability section that explains that.
- 2) Your manuscript contains statistics and error bars based on $n=2$. Please use scatter blots in these cases. No statistics should be calculated if $n=2$.

When submitting your revised manuscript, please carefully review the instructions that follow below. Failure to include requested items will delay the evaluation of your revision.*****

- 1) a .docx formatted version of the manuscript text (including legends for main figures, EV figures and tables). Please make sure that the changes are highlighted to be clearly visible.
- 2) individual production quality figure files as .eps, .tif, .jpg (one file per figure). Please download our Figure Preparation Guidelines (figure preparation pdf) from our Author Guidelines pages <https://www.embopress.org/page/journal/14693178/authorguide> for more info on how to prepare your figures.
- 3) a .docx formatted letter INCLUDING the reviewers' reports and your detailed point-by-point responses to their comments. As part of the EMBO Press transparent editorial process, the point-by-point response is part of the Review Process File (RPF), which will be published alongside your paper.
- 4) a complete author checklist, which you can download from our author guidelines (<<https://www.embopress.org/page/journal/14693178/authorguide>>). Please insert information in the checklist that is also

reflected in the manuscript. The completed author checklist will also be part of the RPF.

5) Please note that all corresponding authors are required to supply an ORCID ID for their name upon submission of a revised manuscript (<<https://orcid.org/>>). Please find instructions on how to link your ORCID ID to your account in our manuscript tracking system in our Author guidelines (<<https://www.embopress.org/page/journal/14693178/authorguide#authorshipguidelines>>)

6) We replaced Supplementary Information with Expanded View (EV) Figures and Tables that are collapsible/expandable online. A maximum of 5 EV Figures can be typeset. EV Figures should be cited as "Figure EV1, Figure EV2" etc... in the text and their respective legends should be included in the main text after the legends of regular figures.

7) "Data Availability" section: the suggested wording is: "The [structural coordinates | microarray | mass spectrometry] data from this publication have been deposited to the [name of the database] database [URL] and assigned the identifier [accession | permalink | hashtag]."). We also need links that resolve directly to a page where the data can be accessed, i.e., directly to the dataset and not just the database.

Additional information on source data and instruction on how to label the files are available <<https://www.embopress.org/page/journal/14693178/authorguide#sourcedata>>.

10) Figure legends and data quantification:
The following points must be specified in each figure legend:

- the name of the statistical test used to generate error bars and P values,
- the number (n) of independent experiments (please specify technical or biological replicates) underlying each data point,
- the nature of the bars and error bars (s.d., s.e.m.)

- If the data are obtained from n {less than or equal to} 5, show the individual data points in addition to the SD or SEM.
- If the data are obtained from n {less than or equal to} 2, use scatter blots showing the individual data points.

See also the guidelines for figure legend preparation:
<https://www.embopress.org/page/journal/14693178/authorguide#figureformat>

11) Our journal encourages inclusion of *data citations in the reference list* to directly cite datasets that were re-used and obtained from public databases. Data citations in the article text are distinct from normal bibliographical citations and should directly link to the database records from which the data can be accessed. In the main text, data citations are formatted as follows: "Data ref: Smith et al, 2001" or "Data ref: NCBI Sequence Read Archive PRJNA342805, 2017". In the Reference list, data citations must be labeled with "[DATASET]". A data reference must provide the database name, accession number/identifiers and a resolvable link to the landing page from which the data can be accessed at the end of the reference. Further instructions are available at <<https://www.embopress.org/page/journal/14693178/authorguide#referencesformat>>.

12) All Materials and Methods need to be described in the main text using our 'Structured Methods' format. According to this format, the Methods section includes a Reagents and Tools Table (listing key reagents, experimental models, software and

relevant equipment and including their sources and relevant identifiers) followed by a Methods and Protocols section describing the methods, ideally using a step-by-step protocol format. The aim is to facilitate adoption of the methodologies across labs. Please download and fill our Reagents and Tools Table template (.docx), which you can find in our author guidelines: <https://www.embopress.org/page/journal/14693178/authorguide#structuredmethods>. When submitting your revised manuscript, please do not include the Reagents and Tools Table in the Methods section of the manuscript but upload it as a separate file choosing the file type "Reagent Table". An example of a Method paper with Structured Methods can be found here: <https://www.embopress.org/doi/10.15252/msb.20178071>.

13) As part of the EMBO publication's Transparent Editorial Process, EMBO Reports publishes online a Review Process File to accompany accepted manuscripts. This File will be published in conjunction with your paper and will include the referee reports, your point-by-point response and all pertinent correspondence relating to the manuscript.

Yours sincerely,

=====

Referee #1:

In this study, Li et al. report that the heterogenous expression of downstream effectors of the Hippo pathway, namely, YAP/TAZ leads to protection against ferroptosis, partly due to the ability of YAP/TAZ-low expressing cells to protect YAP/TAZ-high cells. The authors reported that cells expressing low YAP/TAZ upregulate GTP cyclohydrolase 1 (GCH1) that drives the synthesis of the antioxidant tetrahydrobiopterin (BH4). Accordingly, BH4 prevents lipid peroxidation to suppress ferroptosis. It is an interesting and clearly laid out study. However, the study falls short in some instances, especially in rigor. It does not explain how the YAP/TAZ-low cells are protected or would respond when challenged by other ferroptosis inducers, including cysteine deprivation, erastin or BSO. It is also unclear if the YAP/TAZ high cells were taking up BH4 and, importantly, the study relies almost exclusively on one cell line thus failing in rigor. The absence of metabolomics or other biochemical assays assessing the levels of reactive oxygen species (ROS) makes it hard to determine if other potential players in ferroptosis are also at play. The study is too limited in scope and would benefit from additional experiments.

Major

1. Given that the premise of this work is that tumors have a heterogenous expression of YAP/TAZ, it would have been nice to see that the authors screened multiple lung cancer cell lines for the expression of YAP/TAZ to justify the selection of at least two cells with distinct expression for further studies. In other words, the observation should be tested in at least one other cell line.
2. Publicly available datasets should be leveraged to analyze survival outcome in YAP/TAZ high vs low tumors. In addition, correlation between YAP/TAZ and GCH1 should also be tested at gene level.
3. Although measurement of BH4 was shown, that was intracellular. Extracellular BH4 should be tested in both YAP/TAZ high/low cells. Similarly, one would expect ROS level to be different in YAP/TAZ low vs high cells and ROS following BH4 supplementation.
4. Metabolomics profiling would be important in assessing the profile of amino acids, including the likes of cysteine and methionine that can be linked to ferroptosis. In the current state, focusing on only BH4 is too narrow in scope even though this is justified by the observation of high GCH1.
5. Ferroptosis inducers beyond RSL3 should be tested in the YAP/TAZ high low cells.

Minor

1. Update the legends to include the concentration of the drugs used in experiments.
2. It would make sense to make clear that this study is focused on lung cancer. This should be reflected on the title and abstract.

Referee #2:

In this study, Li et al. report tetrahydrobiopterin (BH4) as a secretory metabolite negatively regulated by YAP/TAZ, which is crucial for YAP/TAZ deficiency-induced resistance to ferroptosis. YAP/TAZ controls BH4 production by transcriptionally inhibiting GTP cyclohydrolase 1 (GCH1), an enzyme required for BH4 biosynthesis. YAP/TAZ knockout cells protected surrounding wildtype cells from ferroptosis via the GCH1-BH4 metabolic axis. Moreover, clinical sample analyses suggest that heterogeneous YAP/TAZ distribution correlates with poor survival rates of cancer patients. Although this study provides a new mechanism by which YAP/TAZ regulate ferroptosis, it lacks clear pathological evidence to confirm whether the findings from cell lines can be truly reflected in vivo.

Major issues

1. Regarding the in vivo findings of heterogeneous YAP/TAZ expression (Fig 1A-C), the following points should be clearly addressed.

- 1) Do the cells used to compare high- and low-YAP/TAZ come from the same tissue? If so, will other types of cells in microenvironment with varying YAP/TAZ expression also influence tumor sensitivity to ferroptosis?
- 2) How were homogeneous and heterogeneous YAP/TAZ expressions defined for the patient survival study? Why were recurrence-free and cancer specific survivals selected for the analysis?
- 3) The authors state that YAP/TAZ expression heterogeneity was observed in both in situ and invasive LUAD samples (Fig 1A) (line 119-120). However, the overall expression levels of YAP/TAZ in these two samples differ significantly. The YAP/TAZ expression level considered "low" in in situ LUAD cells was comparable to that considered "high" in invasive LUAD cells. Based on the proposed model, does this imply that similarly expressed YAP/TAZ control CGH1 expression and BH4 production differently in these two LUAD tissues?
- 4) The authors should explain how such YAP/TAZ heterogeneity arises within the same tumor tissue. Does this heterogeneity evolve during tumorigenesis and tumor progression? In individual tumor cells, does YAP/TAZ expression remain stable or fluctuate?
- 5) What role does YAP/TAZ heterogeneity play in normal tissues and physiology, including development, regeneration, and organ size, by regulating resistance to ferroptosis?
- 6) How is ferroptosis-caused cell death induced in tumors?

2. The discovered GCH1-BH4 metabolic axis should be validated in patient samples, considering YAP/TAZ heterogeneity. Inferring the in vivo situation solely based on the findings from YAP/TAZ dKO cells is problematic and may not accurately reflect the complexity of YAP/TAZ heterogeneity in a clinical context.

3. Since only one cell line and one ferroptosis compound were used throughout the entire study, there is concern regarding specificity issue. The authors should validate their findings in different cell lines, tissues, and with different ferroptosis compounds.

4. In Fig 2H, authors should confirm GCH1 gene is a direct transcriptional target of the YAP/TAZ-TEAD complex through ChIP assay and explore how YAP/TAZ inhibit GCH1 transcription.

Minor point

In Fig 1E, why was the non-specific band level also affected by YAP/TAZ?

Referee #3:

Cellular heterogeneity influences cancer progression and treatment outcomes. Here, Li et al demonstrate that the Hippo pathway, a key regulator of cell growth and survival, plays a crucial role in this context by activating the proteins YAP and TAZ. Using human lung adenocarcinoma and murine models, the authors report varying levels of YAP/TAZ activity, especially in advanced tumor stages. Cells with lower YAP/TAZ activity grow slowly but resist ferroptosis, while cells with higher YAP/TAZ activity grow rapidly but are more vulnerable to ferroptosis. Remarkably, low YAP/TAZ activity cells can protect neighboring cells from ferroptosis, creating a microenvironment that enhances overall tumor resistance. This protective effect is driven by the upregulation of GTP cyclohydrolase 1 (GCH1), which produces an antioxidant, BH4, to prevent lipid peroxidation. Importantly, inhibiting GCH1 sensitizes tumors to ferroptosis-inducing treatments, offering a promising new therapeutic avenue. Overall, this study sheds light on the complex interactions between cancer cells with varying Hippo pathway activities, and the potential of targeting GCH1 to improve ferroptosis based strategies. The work is well performed and of interest. I only have a few points the authors might wish to address:

1- Please provide a full dose-response curve for RSL3 in the YAP/TAZ double KO cells. This is important to understand the level of protection provided by its loss.

2- Western blot misses the MW; please add.

3- Line 195 - the authors mention data not shown, I am not a big fan of such statement and would encourage the authors to add this information to the manuscript.

4- An aspect of the work that the authors do not address or discuss is the identity of the BH3/BH4 uptake mechanism by cells. It appears that this metabolite is transported by the equilibrative transporters ENT1 and ENT2. While not obligatory, addressing whether the deletion of these transporters could blunt the protective effect of co-culture could provide unambiguous evidence for the proposed mechanism.

Response to Reviewer #1

In this study, Li et al. report that the heterogenous expression of downstream effectors of the Hippo pathway, namely, YAP/TAZ leads to protection against ferroptosis, partly due to the ability of YAP/TAZ-low expressing cells to protect YAP/TAZ-high cells. The authors reported that cells expressing low YAP/TAZ upregulate GTP cyclohydrolase 1 (GCH1) that drives the synthesis of the antioxidant tetrahydrobiopterin (BH4). Accordingly, BH4 prevents lipid peroxidation to suppress ferroptosis. It is an interesting and clearly laid out study. However, the study falls short in some instances, especially in rigor. It does not explain how the YAP/TAZ-low cells are protected or would respond when challenged by other ferroptosis inducers, including cysteine deprivation, erastin or BSO. It is also unclear if the YAP/TAZ high cells were taking up BH4 and, importantly, the study relies almost exclusively on one cell line thus failing in rigor. The absence of metabolomics or other biochemical assays assessing the levels of reactive oxygen species (ROS) makes it hard to determine if other potential players in ferroptosis are also at play. The study is too limited in scope and would benefit from additional experiments.

We thank the reviewer for their positive opinion of our study and for the constructive comments to improve the quality of this study. The specific responses to the points raised are as follows:

Major

1. Given that the premise of this work is that tumors have a heterogenous expression of YAP/TAZ, it would have been nice to see that the authors screened multiple lung cancer cell lines for the expression of YAP/TAZ to justify the selection of at least two cells with distinct expression for further studies. In other words, the observation should be tested in at least one other cell line.

(Response) To validate our findings in other lung cancer cell lines, we evaluated the expression and activity of YAP/TAZ by Phos-tag immunoblotting in LK-2, PC-9, H226 and H358 human lung cancer cells. Since LK-2 and H226 cells exhibited higher YAP activity (lower phosphorylation levels of YAP) than the other two cell lines (**Data provided for reviewer's information, Fig A**), we selected these two cell lines to confirm the effect of YAP/TAZ ablation on ferroptosis sensitivity. While we were unable to establish YAP/TAZ-deficient H226 cells due to toxicity upon YAP/TAZ removal, we successfully established YAP/TAZ-deficient LK-2 cells. Similar to the results observed in LLC cells, YAP/TAZ depletion suppressed ferroptosis in LK-2 cells. We have included these data (**New Fig EV3E and EV3F**) along with its interpretation in the Results section (**page 7, lines 183 - 184**).

Data provided for reviewer's information: *Expression and phosphorylation of YAP/TAZ in human lung cancer cell lines.*

(A) Immunoblotting (IB) of cell extracts from LK-2, PC-9, H226 and H358 cells with antibodies against indicated proteins was shown.

2. Publicly available datasets should be leveraged to analyze survival outcome in YAP/TAZ high vs low tumors. In addition, correlation between YAP/TAZ and GCH1 should also be tested at gene level.

(Response) We thank the reviewer for this suggestion. Regarding the survival outcome in YAP/TAZ high vs. low tumors, we propose that YAP/TAZ heterogeneity within the same tumor tissue is associated with poor prognosis in patients with lung adenocarcinoma. Therefore, the scope of this study is not focused on the difference between lung cancers with high YAP/TAZ expression and those with low YAP/TAZ expression. Thus, we have not explored the first point. Regarding the second point (the correlation between YAP/TAZ and GCH1), we examined the association between GCH1 expression and YAP expression, as well as YAP target genes (to better monitor its activity), using the GEPIA (Gene Expression Profiling Interactive Analysis) database. We found that GCH1 expression is negatively correlated with both YAP expression and YAP target genes in human tissues. Accordingly, we have included these data (**New Fig EV4D and EV4E**) along with its interpretation in the Results section (**page 8, lines 234 - 237**).

3. Although measurement of BH4 was shown, that was intracellular. Extracellular BH4 should be tested in both YAP/TAZ high/low cells. Similarly, one would expect ROS level to be different in YAP/TAZ low vs high cells and ROS following BH4 supplementation.

(Response) We thank the reviewer for this important comment. As BH4 is easily oxidized to BH2 outside of the cell (Cunnington *et al*, 2012), we evaluated extracellular BH2 levels for this purpose. Indeed, we were unable to detect BH4 in the media despite the large number of cells cultured to collect conditioned media. Instead, we found that extracellular BH2 levels were markedly higher in YAP/TAZ dKO cells than in WT cells. We have included these data (**New Fig EV5B**) along with their interpretation in the Results section (**page 9, lines 256 - 258**). Regarding ROS levels in WT and YAP/TAZ dKO cells following BH4 supplementation, we have already provided the results evaluating the lipid peroxide levels in these cells in the presence or absence of BH4 in **Original Fig EV4D (New Fig EV5E)**.

4. Metabolomics profiling would be important in assessing the profile of amino acids, including the likes of cysteine and methionine that can be linked to ferroptosis. In the current state, focusing on only BH4 is too narrow in scope even though this is justified by the observation of high GCH1.

(Response) We thank the reviewer for raising this issue. Because cysteine and methionine are the precursor for glutathione (GSH) biosynthesis, we measured intracellular GSH levels in WT and YAP/TAZ dKO cells. The amount of GSH was comparable between the two cell lines, negating the contribution of cysteine and methionine metabolism as well as GSH metabolism to ferroptosis resistance elicited by YAP/TAZ depletion in LLC cells. Taken these new results together with other data in our manuscript, we concluded that upregulation of GCH1-mediated BH4 production is a key mechanism conferring ferroptosis resistance in YAP/TAZ dKO cells. We have included the aforementioned data (**New Fig EV4B**) along with its interpretation in the Results section (**page 8, lines 2210- 224**).

5. Ferroptosis inducers beyond RSL3 should be tested in the YAP/TAZ high low cells.

(Response) According to the reviewer's suggestion, we examined the effect of YAP/TAZ depletion on ferroptosis induced by Erastin, L-Buthionine-(S,R)-Sulfoximine (L-BSO), cysteine and cystine starvation, or Sulfasalazine (SAS). YAP/TAZ depletion suppressed ferroptosis induced by these conditions. We have included the aforementioned data (**New Fig EV2H and EV2I**) along with its interpretation in Results section (**page 6, lines 166 - 172**).

Minor

1. Update the legends to include the concentration of the drugs used in experiments.

(Response) We thank the reviewer for this suggestion and have carefully revised the manuscript accordingly.

2. It would make sense to make clear that this study is focused on lung cancer. This should be reflected on the title and abstract.

(Response) We agree with the reviewer and have now revised the manuscript accordingly.

Response to Reviewer #2

In this study, Li et al. report tetrahydrobiopterin (BH4) as a secretory metabolite negatively regulated by YAP/TAZ, which is crucial for YAP/TAZ deficiency-induced resistance to ferroptosis. YAP/TAZ controls BH4 production by transcriptionally inhibiting GTP cyclohydrolase 1 (GCHI), an enzyme required for BH4 biosynthesis. YAP/TAZ knockout cells protected surrounding wildtype cells from ferroptosis via the GCHI-BH4 metabolic axis. Moreover, clinical sample analyses suggest that heterogeneous YAP/TAZ distribution correlates with poor survival rates of cancer patients. Although this study provides a new mechanism by which YAP/TAZ regulate ferroptosis, it lacks clear pathological evidence to confirm whether the findings from cell lines can be truly reflected in vivo.

We thank the reviewer for their constructive comments, which have greatly contributed to improving the quality of this study. The specific responses to the points raised are as follows:

Major issues

1. Regarding the in vivo findings of heterogeneous YAP/TAZ expression (Fig 1A-C), the following points should be clearly addressed.

1) Do the cells used to compare high- and low-YAP/TAZ come from the same tissue? If so, will other types of cells in microenvironment with varying YAP/TAZ expression also influence tumor sensitivity to ferroptosis?

(Response) We apologize for the insufficient explanation of the pathological data in the original manuscript. In this study, we examined the expression pattern of YAP/TAZ in cancer cells within the same lung adenocarcinoma tissues. In the original manuscript, cancer cells were distinguished from other stromal cells based on their morphological features by pathologists. To validate our pathological evaluation of cancer cells, we performed consecutive immunostaining for YAP/TAZ and a cancer cell marker CK (AE1/AE3) in the representative sections of lung adenocarcinoma tissues. Consistent with our initial investigation, CK⁺ cancer cells exhibited both homogenous and heterogeneous expression patterns of YAP/TAZ within tumor tissues. Notably, YAP/TAZ were predominantly expressed in cancer cells within lung adenocarcinoma tissues. Based on these observations, we concluded that varying YAP/TAZ expression in cancer cells is associated with poor patient survival. We have included the aforementioned data (**New Fig 1B**) along with its interpretation in Results section (**page 5, lines 126 - 128**).

2) How were homogeneous and heterogeneous YAP/TAZ expressions defined for the patient survival study? Why were recurrence-free and cancer specific survivals selected for the analysis?

(Response) We classified cancer cells into YAP/TAZ^{high} (positive) and YAP/TAZ^{low} (negative) cells according to the nuclear levels of YAP/TAZ. Cases with more than 80% of cancer cells exhibiting YAP/TAZ^{high} expression were classified as the “homogenous” group, while cases with less than 80% of cancer cells exhibiting YAP/TAZ^{high} expression were classified as the “heterogeneous” group. In this study, we selected recurrence-free survival (RFS) and cancer specific survival (CSS) as the primary outcomes for the following reasons:

1. Postoperative recurrence is common in lung cancer and has a significant impact on prognosis.
2. To evaluate the direct impact of cancer on prognosis, we selected CSS rather than overall survival.

We have revised the manuscript to clarify the definitions of “homogenous” and “heterogeneous” groups in the Materials and Methods section (**page 22, lines 678 - 682**).

3) The authors state that YAP/TAZ expression heterogeneity was observed in both in situ and

invasive LUAD samples (Fig 1A) (line 119-120). However, the overall expression levels of YAP/TAZ in these two samples differ significantly. The YAP/TAZ expression level considered "low" in in situ LUAD cells was comparable to that considered "high" in invasive LUAD cells. Based on the proposed model, does this imply that similarly expressed YAP/TAZ control CGHI expression and BH4 production differently in these two LUAD tissues?

(Response) We apologize for the misleading images presented in the original manuscript, where we did not pay sufficient attention to the staining intensity of the YAP/TAZ immunoreactive signal. Our aim was to highlight the heterogeneity observed in lung adenocarcinoma samples. As mentioned earlier, the classification of "homogeneous" and "heterogeneous" expression patterns was not based on the total amount of YAP/TAZ expression within the patient group but rather on the intratumoral heterogeneity within the same tumor samples. We carefully reviewed the images of tumor tissues and confirmed that there was no apparent difference in the overall expression of YAP/TAZ between the in situ and invasive groups. Therefore, in the current version of the manuscript, we have replaced the representative images of "homogeneous" and "heterogeneous" expression patterns of YAP/TAZ staining to avoid any misunderstanding (**New Fig 1A**). We sincerely thank the reviewer for this important comment.

4) The authors should explain how such YAP/TAZ heterogeneity arises within the same tumor tissue. Does this heterogeneity evolve during tumorigenesis and tumor progression? In individual tumor cells, does YAP/TAZ expression remain stable or fluctuate?

(Response) Although we agree with the reviewer that the mechanisms underlying the genesis of YAP/TAZ heterogeneity are intriguing, to the best of our knowledge, no study has clearly elucidated a mechanism explaining the emergence of YAP/TAZ heterogeneity in tumors or normal tissues. Since the Hippo pathway plays a crucial role in responding to various intracellular and extracellular stimuli, including soluble factors, mechanical signals, and nutrient signals (Hansen *et al*, 2015), we speculate that local environmental factors within the tumor microenvironment may influence YAP/TAZ activation, contributing to the observed heterogeneity. We believe that identifying the molecular basis for the development of YAP/TAZ heterogeneity in cancer cells during the progression of cancers is an important area for future studies. Accordingly, we have added these points to the Discussion section to emphasize the significance of this issue (**page 13, lines 397 - 399**).

5) What role does YAP/TAZ heterogeneity play in normal tissues and physiology, including development, regeneration, and organ size, by regulating resistance to ferroptosis?

(Response) Recent studies have suggested the role of YAP/TAZ heterogeneity in mammalian development (Nita & Moroishi, 2024). At the mid-blastocyst stage, there is considerable variability in nuclear YAP activity and the expression of pluripotency factors within the inner cell mass. As epiblast formation progresses, YAP gradually accumulates in the nucleus, triggering the activation of TEAD-driven transcriptional programs. The increased TEAD transcriptional activity promotes the expression of pluripotency factors, which are crucial for epiblast formation. During this process, cells compete to eliminate those with low TEAD activity via apoptosis, effectively removing less pluripotent cells. This mechanism ensures the generation of high-quality epiblasts, which are characterized by a state of naïve pluripotency (Hashimoto & Sasaki, 2020). Given the recently reported physiological role of ferroptosis in avian muscle tissue sculpting (Co *et al*, 2024), it would be an intriguing future direction to explore whether ferroptosis regulation by YAP/TAZ heterogeneity contributes to organ development. Accordingly, we added these points in the Discussion section (**page 13, lines 385 - 397**).

6) How is ferroptosis-caused cell death induced in tumors?

(Response) Cancer cell ferroptosis has recently gained attention as a potential novel treatment strategy. Numerous studies have investigated the therapeutic effects of drugs targeting ferroptosis-related molecules such as GPX4 and xCT on various types of cancers. In addition to the therapeutic induction of ferroptosis, accumulating evidence has suggested that cancer cells encounter pro-ferroptotic stimuli within the tumor microenvironment or during metastasis. For instance, CD8⁺ T cells suppress xCT expression in cancer cells via IFN γ , thereby sensitizing them to ferroptosis (Wang *et al*, 2019). Another study proposed that metastasizing melanoma cells are subjected to more ferroptosis stress in the blood than in the lymph, which may explain why metastasis through the blood is less efficient compared to lymphatic metastasis (Ubellacker *et al*, 2020). Accordingly, we added these points in the Discussion section (**page 12, lines 343 - 352**).

2. The discovered GCH1-BH4 metabolic axis should be validated in patient samples, considering YAP/TAZ heterogeneity. Inferring the in vivo situation solely based on the findings from YAP/TAZ dKO cells is problematic and may not accurately reflect the complexity of YAP/TAZ heterogeneity in a clinical context.

(Response) We thank the reviewer for raising this issue. To address the reviewer's points, histological evaluation of GCH1 expression in patient samples would be required. However, to the best of our knowledge, there is no commercially available antibody against human GCH1 that has been validated for immunohistochemistry. Therefore, we tested the GCH1 antibody (#PA5-120098; Thermo Fisher Scientific) used in this study to validate its suitability for immunostaining experiments. We stained WT (low GCH1 expression), YAP/TAZ dKO (high GCH1 expression) and YAP/TAZ/GCH1 tKO (no GCH1 expression) LLC cells with this antibody. We observed comparable signal among these three cell lines (Figure A), indicating that the current GCH1 antibody is not suitable for immunostaining experiments, although it can be used for Western blotting, as shown in this study. Consequently, we were unable to validate the link between YAP/TAZ heterogeneity and GCH1 expression in patient samples. We agree with the reviewer's points and have revised the discussion to highlight the potential limitations of this study (**page 12 - 13, lines 372 - 373**).

Data provided for reviewer's information: *The commercially available GCH1 antibody (#PA5-120098; Thermo Fisher Scientific) is not suitable for immunostaining.*

(A) WT, YAP/TAZ dKO and YAP/TAZ/GCH1 tKO LLC cells showed non-specific staining when they were subjected to immunostaining with the GCH1 antibody.

3. Since only one cell line and one ferroptosis compound were used throughout the entire study, there is concern regarding specificity issue. The authors should validate their findings in different cell lines, tissues, and with different ferroptosis compounds.

(Response) In response to the reviewer's suggestion, we examined the effect of YAP/TAZ depletion on ferroptosis induced by Erastin, L-Buthionine-(S,R)-Sulfoximine (L-BSO), cysteine and cystine starvation (CCS), and Sulfasalazine (SAS). YAP/TAZ depletion also suppressed ferroptosis under these conditions. We have included these data (**New Fig EV2H and EV2I**) along with its interpretation in the Results section (**page 6, lines 166 - 172**). To verify our findings in cell lines other than LLC, we deleted YAP/TAZ in LK-2 human lung cancer cells. Similar to the results observed in LLC cells, YAP/TAZ depletion suppressed ferroptosis in LK-2 cells. We have included these data (**New Fig EV3E and EV3F**) along with its interpretation in the Results section (**page 7, lines 183 - 184**). Since the focus of our study is on lung cancer, we did not validate our findings in other tissues. We have carefully revised the manuscript to clarify this point, reflecting the change in both the title and abstract.

4. In Fig 2H, authors should confirm GCH1 gene is a direct transcriptional target of the YAP/TAZ-TEAD complex through ChIP assay and explore how YAP/TAZ inhibit GCH1 transcription.

(Response) Since YAP/TAZ depletion increased chromatin accessibility at approximately +1 kb from the transcription start site (TSS) of the *Gch1* locus (**Original Fig 2H [New Fig 3H]**), we hypothesized that the YAP/TAZ-TEAD complex suppresses *Gch1* transcription via binding to this region. To examine the enrichment of TEAD1 within the TSS-proximal region, a chromatin immunoprecipitation (ChIP) assay was performed. While apparent binding of TEAD1 was observed at the promoter region of *Ctcf*, a direct target of the YAP/TAZ-TEAD complex used as a positive control, the enrichment of TEAD1 within the TSS proximal region of the *Gch1* gene was negligible. These data suggest that GCH1 may not be a direct transcriptional target of the YAP/TAZ-TEAD complex.

We next investigated the involvement of PBX1 and CREB, transcription factors previously reported to upregulate *Gch1* transcription (Kapatos *et al*, 2007; Liu *et al*, 2022). First, to investigate the role of PBX1, we depleted PBX1 using CRISPR/Cas9 system in YAP/TAZ dKO cells. DNA mutation in the *Pbx1* locus was confirmed by DNA sequencing (Figure B). Additionally, PBX1 depletion reduced mRNA levels of *Nfe2l1*, a known target gene of PBX1, confirming the functional loss of PBX1 in the established YAP/TAZ/PBX1 tKO cells (Figure C). As assessed by immunoblotting, PBX1 depletion did not affect GCH1 expression in YAP/TAZ dKO cells, suggesting that PBX1 may not be involved in the upregulation of GCH1 expression in YAP/TAZ dKO cells. Second, to investigate the role of CREB, we examined CREB phosphorylation in WT and YAP/TAZ dKO cells, as the transcriptional activity of CREB is mainly activated through its phosphorylation. There was no difference in the amount of phosphorylated CREB between the two cell lines. Moreover, treatment with IBMX and forskolin, which induce CREB phosphorylation through upregulation of cAMP levels, did not affect GCH1 expression in either cell line (Figure E). Collectively, these results suggest that CREB activity may not be associated with GCH1 expression in LLC cells.

Despite our efforts to elucidate the molecular mechanisms underlying GCH1 upregulation in YAP/TAZ-deficient cells, we were unable to propose a precise mechanism. This remains an important topic for future studies. We appreciate the reviewer's comments and have revised the discussion to highlight the potential limitations of this study (**page 12, lines 371 - 372**).

Data provided for reviewer's information: Exploration of molecular mechanism underlying induction of *GCH1* expression by *YAP/TAZ* depletion.

(A) TEAD1 does not bind to the TSS proximal region of the *Gch1* gene. Binding of TEAD1 to the uniquely open region observed in *YAP/TAZ* dKO cells was examined via chromatin immunoprecipitation assay. Data are represented as the mean \pm SD of triplicates of a representative experiment. ns indicates no significant difference, *** $p < 0.001$, one-way ANOVA, followed by Tukey's multiple comparison test.

(B) DNA mutation in *YAP/TAZ/PBX1* tKO LLC cells using the clustered regularly interspaced short palindromic repeat (CRISPR)/Cas9 system. gDNA, genomic DNA.

(C) Reverse transcription (RT) and real-time PCR analysis of the *PBX1* target gene *Nfe2l1* in WT and *YAP/TAZ/PBX1* tKO LLC cells. Data are means \pm SD of three biologically independent samples from a representative experiment. *** $p < 0.001$, unpaired t-test.

(D) Immunoblotting (IB) of cell extracts from WT, *YAP/TAZ* dKO and *YAP/TAZ/PBX1* tKO LLC cells with antibodies against the indicated proteins.

(E) WT and YAP/TAZ dKO LLC cells were stimulated with IBMX (100 μ M) and Forskorin (20 μ M) for 4 h. Cell lysates were subjected to immunoblotting with the indicated antibodies.

Minor point

In Fig 1E, why was the non-specific band level also affected by YAP/TAZ?

(Response) We thank the reviewer for raising this issue. Because the upper band (interpreted here as “non-specific”) did not disappear in GCH1-deleted cells (YAP/TAZ/GCH1 tKO cells) in **Original Fig 3C (New Fig 4C)**, we classified it as non-GCH1 band. Additionally, YAP/TAZ depletion did not affect the non-specific band level in **Original Fig 2B (New Fig 3B)** and **Original Fig 3C (New Fig 4C)**. Therefore, the alteration of the non-specific band level observed in **Original Fig 2E (New Fig 3E)** is not reproducible. Taken together, we believe that the proteins detected in the upper non-specific band are not related to our findings.

Response to Reviewer #3

Cellular heterogeneity influences cancer progression and treatment outcomes. Here, Li et al demonstrate that the Hippo pathway, a key regulator of cell growth and survival, plays a crucial role in this context by activating the proteins YAP and TAZ. Using human lung adenocarcinoma and murine models, the authors report varying levels of YAP/TAZ activity, especially in advanced tumor stages. Cells with lower YAP/TAZ activity grow slowly but resist ferroptosis, while cells with higher YAP/TAZ activity grow rapidly but are more vulnerable to ferroptosis. Remarkably, low YAP/TAZ activity cells can protect neighboring cells from ferroptosis, creating a microenvironment that enhances overall tumor resistance. This protective effect is driven by the upregulation of GTP cyclohydrolase 1 (GCH1), which produces an antioxidant, BH4, to prevent lipid peroxidation. Importantly, inhibiting GCH1 sensitizes tumors to ferroptosis-inducing treatments, offering a promising new therapeutic avenue.

Overall, this study sheds light on the complex interactions between cancer cells with varying Hippo pathway activities, and the potential of targeting GCH1 to improve ferroptosis based strategies. The work is well performed and of interest.

I only have a few points the authors might wish to address:

We thank the reviewer for the positive opinion of our study and constructive comments to improve the quality of this study. The specific responses to the points raised are as follows:

1- Please provide a full dose-response curve for RSL3 in the YAP/TAZ double KO cells. This is important to understand the level of protection provided by it loss.

(Response) Following the reviewer's suggestion, we have evaluated a full dose-response curve of RSL3 in WT and YAP/TAZ dKO LLC cells (**New Fig EV2G**) and have revised the manuscript accordingly (**page 6, lines 163 - 164**).

2- Western blot misses the MW; please add.

(Response) We have included the molecular weight information for the Western blot data in the revised manuscript.

3- Line 195 - the authors mention data not shown, I am not a big fan of such statement and would encourage the authors to add this information to the manuscript.

(Response) We appreciate the reviewer's suggestion and have now revised the manuscript accordingly (**page 7, lines 205 - 207**), adding the related data (**New Fig EV4A**).

4- An aspect of the work that the authors do not address or discuss is the identity of the BH3/BH4 uptake mechanism by cells. It appears that this metabolite is transported by the equilibrative transporters ENT1 and ENT2. While not obligatory, addressing whether the deletion of these transporters could blunt the protective effect of co-culture could provide unambiguous evidence for the proposed mechanism.

(Response) We thank the reviewer for this suggestion. As pointed out by the reviewer, a previous study has shown that ENT1 and ENT2 act as transporters for BH4 and BH2 (Ohashi *et al*, 2011). To explore the roles of ENT1 and ENT2 in the uptake of BH4 and BH2 in our context, we suppressed the expression of ENT1 and ENT2 using siRNA (Figure A). BH2-mediated protection against ferroptosis was similar between siControl and siENT1/2-transfected cells (Figure B), suggesting that other mechanisms may contribute to the uptake of BH2 and BH4 in LLC cells. Accordingly, we have added a discussion of the molecular mechanisms involved in BH2/BH4 uptake in the Discussion section, in response to the

reviewer's comment (page 13, lines 379 - 383).

Data provided for reviewer's information: *Knockdown of ENT1/ENT2 did not suppress BH2-induced protection against ferroptosis in LLC cells.*

(A) RT and real-time PCR analysis of the *Ent1/Ent2* gene in siControl and siENT1/ENT2 LLC cells. Data are means \pm SD of three biologically independent samples from a representative experiment. *** $p < 0.001$, unpaired t-test.

(B) siControl and siENT1/ENT2 LLC cells were pretreated with or without BH2 (40 μ M) for 30 min, followed by stimulation with RSL3 (400 nM) for 10 h. Dead cells were stained with propidium iodide (PI), and the percentage of the PI-negative live cell population was calculated using the flow cytometer. Data are means \pm SD of three biologically independent samples from a representative experiment. ns indicates no significant difference, one-way ANOVA test followed by Tukey's multiple comparison test.

Reference:

Co HKC, Wu CC, Lee YC, Chen SH (2024) Emergence of large-scale cell death through ferroptotic trigger waves. *Nature* 631: 654-662

Cunnington C, Van Assche T, Shirodaria C, Kyllintreas I, Lindsay AC, Lee JM, Antoniadis C, Margaritis M, Lee R, Cerrato R *et al* (2012) Systemic and vascular oxidation limits the efficacy of oral tetrahydrobiopterin treatment in patients with coronary artery disease. *Circulation* 125: 1356-1366

Hansen CG, Moroishi T, Guan KL (2015) YAP and TAZ: a nexus for Hippo signaling and beyond. *Trends Cell Biol* 25: 499-513

Hashimoto M, Sasaki H (2020) Cell competition controls differentiation in mouse embryos and stem cells. *Curr Opin Cell Biol* 67: 1-8

Kapatos G, Vunnava P, Wu Y (2007) Protein kinase A-dependent recruitment of RNA polymerase II, C/EBP beta and NF-Y to the rat GTP cyclohydrolase I proximal promoter occurs without alterations in histone acetylation. *J Neurochem* 101: 1119-1133

Liu Y, Zhai E, Chen J, Qian Y, Zhao R, Ma Y, Liu J, Huang Z, Cai S, Chen J (2022) m(6) A-mediated regulation of PBX1-GCH1 axis promotes gastric cancer proliferation and metastasis by elevating tetrahydrobiopterin levels. *Cancer Commun (Lond)* 42: 327-344

Nita A, Moroishi T (2024) Hippo pathway in cell-cell communication: emerging roles in development and regeneration. *Inflamm Regen* 44: 18

Ohashi A, Sugawara Y, Mamada K, Harada Y, Sumi T, Anzai N, Aizawa S, Hasegawa H (2011) Membrane transport of sepiapterin and dihydrobiopterin by equilibrative nucleoside transporters: a plausible gateway for the salvage pathway of tetrahydrobiopterin biosynthesis. *Mol Genet Metab* 102: 18-28

Ubellacker JM, Tasdogan A, Ramesh V, Shen B, Mitchell EC, Martin-Sandoval MS, Gu Z, McCormick ML, Durham AB, Spitz DR *et al* (2020) Lymph protects metastasizing melanoma cells from ferroptosis. *Nature* 585: 113-118

Wang W, Green M, Choi JE, Gijón M, Kennedy PD, Johnson JK, Liao P, Lang X, Kryczek I, Sell A *et al* (2019) CD8(+) T cells regulate tumour ferroptosis during cancer immunotherapy. *Nature* 569: 270-274

Dear Toshiro,

Thank you once more for your patience while your manuscript was under review and thank you also for providing feedback on the remaining concerns from referee #1.

As discussed and agreed upon, please provide further Western blots and quantification in response to point (1) and please discuss points (2) and (3) in the manuscript text and point-by-point response. The proposed timeline of 4 weeks is fine, given that you have only recently moved your lab.

From the editorial side, there are also a few things that we need before we can proceed with the official acceptance of your study.

- Please reduce a number of keywords to five.
 - Regarding the Author Contributions, we now use CRediT to specify the contributions of each author in the journal submission system. Therefore, please remove the Author Contributions from the manuscript file and make sure that the author contributions in our online manuscript tracking system are correct and up-to-date. The information you specified in the system will be automatically retrieved and typeset into the article. You can enter additional information in the free text box provided, if you wish.
 - Figure callouts: "Supplementary information" in the text needs to be removed or updated to the correct nomenclature.
 - The legends for the EV Figures need to be moved to the main manuscript text, to a separate section called Expanded View Figure legends, just after the main figure legends. The file on "Supplemental Information" is then not needed anymore.
 - Table EV2 and EV3 should be Datasets EV#. Please add a legend in a separate Tab of the .xls file.
 - The nomenclature of the separate EV figure needs to be changed to Figure EV1, etc. instead of Figure S1, etc. (The name on the figure itself).
 - The manuscript sections should be in the following order: Title page - Abstract & Keywords - Introduction - Results - Discussion - Methods - Data Availability - Acknowledgments - Disclosure Statement & Competing Interests - References - Figure Legends - (Main Tables with legends if applicable) - Expanded View Figure Legends.
 - Materials and Methods should be Methods.
 - The data availability section should only inform about datasets deposited in public repositories. Therefore, please remove the following statement: "Supplementary information is available for this paper. Correspondence and requests for materials should be addressed to Toshiro Moroishi (moroishi.toshiro@tmd.ac.jp)."
 - Author checklist: please complete the information in row 6, Manuscript number.
 - Author checklist: you filled row 58, but please note that this only applies to field work, e.g., animals captured from the field. Please check.
 - Author Checklist, row 101, dual use research of concern: you refer to a statement in the Acknowledgments but I am not sure whether Dual Use Research of Concern (DURC) indeed applies to your manuscript? If it does, we would need information on the authority granting approval and the reference number. Please clarify.
<https://www.selectagents.gov/sat/list.htm>
 - Mouse experimental procedures: please also provide the approval reference number in addition to the approving committee in the methods.
 - Please upload the Source Data as one folder per figure with subfolders for the individual panels.
 - Source Data for Figure 1B seems missing.
 - Our production/data editors have asked you to clarify several points in the figure legends (see below). Please incorporate these changes in the manuscript and return the revised file with tracked changes with your final manuscript submission.
- A) Statistical test information. Only p-values that are actually shown in the figure panel(s) should (and must) be defined in the legends, all others should be removed from (or added to) the legend. Moreover, we ask for the specification of exact p-values:
- Please note that the exact p values are not provided in the legends of figures 1C, 2C; 3C, D, F, G; 4B, D, E, F; 5A, D, E; EV1 C, EV2 A-F, H, I; EV3 A, C, D, F; EV4 B, C; EV5 A, B, D, E.

- Please indicate the statistical test used for data analysis in the legend of figure 3A

B) Replicates and error bars:

- Please note that information related to n is missing in the legend of figure 3A
- Please note that the error bars are not defined in the legends of figures EV2 G; EV4 A

D) Data presentation:

- Please note that the white arrow heads are not defined in the legend of figure 1B. This needs to be rectified.

- As a standard procedure, we edit the title and abstract of manuscripts to make them more accessible to a general readership. Please find the suggested versions below my signature.

- Finally, EMBO Reports papers are accompanied online by

A) a short (1-2 sentences) summary of the findings and their significance,

B) 2-3 bullet points highlighting key results and

C) a schematic summary figure that provides a sketch of the major findings (not a data image).

Please provide the summary figure as a separate file in PNG or JPG format at a size of 550x300-600 pixels (width x height).

Please note that the size is rather small and that text needs to be readable at the final size. Please send us this information along with the revised manuscript.

With kind regards,

=====

Referee #1:

Thank you to the authors for the rebuttals. However, I still have a couple of points:

1. The western blot data provided in response to the review point #1 should be included in the manuscript main Figure, preferably in the main Figure 1. That said, it is hard to tell from that western that the phospho-YAP is much different between the cells. Looking at the actin, it is fair to assume that PC-9, for example, had more sample loading than LK-2. In such a case where the result is not clear, especially for basal comparison where no complicated experiment is warranted, the western blot should be repeated, quantified and the representative blot + quantification presented.

2. The response to review point #2, namely, "...the scope of this study is not focused on the difference between lung cancers with high YAP/TAZ expression and those with low YAP/TAZ expression. Thus, we have not explored the first point" is not satisfactory. Granted that a high vs low YAP/TAZ comparison is not directly aligned with the authors' proposition of heterogeneity, there has to be a way to show that a more heterogenous YAP/TAZ expression correlates with survival outcome. For example, the authors could compare the 1st and 3rd quartile of a lung cancer cohort (representing the lowest/highest YAP/TAZ expressing groups) with the rest of the data in the cohort. The TCGA data and related cohorts should enable this comparison.

3. The ferroptosis inducer data presented as part of rebuttal to point #5 should be done for at least PC-9 (that cell is being suggested based on the basal western blot data). It may be worthwhile, although optional, to do YAP/TAZ knockout any of those additional cells, but the basal validation of differential response to the ferroptosis inducers should be done and typically can be delivered within a very short time frame. If YAP is indeed different in the cells, this experiment should reflect differences in the way the cells respond to the inducers.

Referee #2:

I appreciate the authors' efforts in performing additional experiments and providing further clarifications and discussions, which have addressed my previous critiques. Therefore, I support the acceptance of this manuscript for publication in EMBO Reports.

Referee #3:

The authors have sufficiently addressed my points. I have no additional comments.

=====

Hippo pathway controls biopterin metabolism to shield adjacent cells from ferroptosis in lung cancer

Recent advances in single-cell technologies have uncovered significant cellular diversity in tumors, influencing cancer progression and treatment outcomes. The Hippo pathway controls cell proliferation through its downstream effectors: yes-associated protein (YAP) and transcriptional co-activator with PDZ-binding motif (TAZ). Our analysis of human lung adenocarcinoma and murine models reveals that cancer cells display heterogeneous YAP/TAZ activation levels within tumors. Murine lung cancer cells with high YAP/TAZ activity grow rapidly but are sensitive to ferroptosis, a cell death induced by lipid peroxidation. In contrast, cells with low YAP/TAZ activity grow slowly but resist ferroptosis. Moreover, they protect neighbouring cells from ferroptosis, creating a protective microenvironment that enhances the tumor's resistance to ferroptosis. Mechanistically, inhibiting YAP/TAZ upregulates GTP cyclohydrolase 1 (GCH1), an enzyme critical for the biosynthesis of tetrahydrobiopterin (BH4), which functions as a secretory antioxidant to prevent lipid peroxidation. Pharmacological inhibition of GCH1 sensitizes lung cancer cells to ferroptosis inducers, suggesting a potential therapeutic approach. Our data highlights the non-cell-autonomous roles of the Hippo pathway in creating a ferroptosis-resistant tumor microenvironment.

Response to Referee #1:

Thank you to the authors for the rebuttals. However, I still have a couple of points:

We thank the reviewer for their constructive comments to improve the quality of this study. The specific responses to the points raised are as follows:

1. The western blot data provided in response to the review point #1 should be included in the manuscript main Figure, preferably in the main Figure 1. That said, it is hard to tell from that western that the phospho-YAP is much different between the cells. Looking at the actin, it is fair to assume that PC-9, for example, had more sample loading than LK-2. In such a case where the result is not clear, especially for basal comparison where no complicated experiment is warranted, the western blot should be repeated, quantified and the representative blot + quantification presented.

(Response) We apologize for the insufficient explanation of our approach to evaluating YAP phosphorylation status. In the previous round of revision, we provided Phos-tag Western blot data rather than phospho-YAP immunoblotting. The Phos-tag method separates YAP proteins based on their phosphorylation status, with more heavily phosphorylated forms migrating more slowly (DOI: 10.1016/j.xpro.2021.101102). This technique allows for a relative assessment of phosphorylation status within individual samples, independent of total protein loading. The ratio of non-phosphorylated YAP (bottom band) to fully phosphorylated YAP (top band) was used to estimate the relative activity of YAP in each cell line. To further support our findings, we have repeated the Phos-tag Western blot experiments and included representative blot images along with quantitative analysis of three biologically independent samples in the **new Fig EV3E**. As these data provide a rationale for selecting LK-2 cells for the validation of YAP/TAZ deletion, we have included them immediately before Figs EV3F and EV3G (**lines 184 - 191**). We thank the reviewer for their insightful suggestion.

2. The response to review point #2, namely, "...the scope of this study is not focused on the difference between lung cancers with high YAP/TAZ expression and those with low YAP/TAZ expression. Thus, we have not explored the first point" is not satisfactory. Granted that a high vs low YAP/TAZ comparison is not directly aligned with the authors' proposition of heterogeneity, there has to be a way to show that a more heterogenous YAP/TAZ expression correlates with survival outcome. For example, the authors could compare the 1st and 3rd quartile of a lung cancer cohort (representing the lowest/highest YAP/TAZ expressing groups) with the rest of the data in the cohort. The TCGA data and related cohorts should enable this comparison.

(Response) We apologize for the insufficient explanation in the previous revision round. While we appreciate the reviewer's suggestion to use publicly available datasets (e.g., TCGA) to correlate lowest/highest YAP/TAZ expression levels with patient outcomes, we are concerned that this approach does not align with the core hypothesis of our study, which focuses on intra-tumoral heterogeneity of YAP/TAZ expression. Publicly available datasets such as TCGA are based on bulk RNA-sequencing, which captures the average gene expression across entire tumor tissues, including both cancer and stromal cells. This method lacks the resolution to detect single-cell-level heterogeneity, which is central to our hypothesis.

For example, the following two hypothetical samples would appear identical in bulk data despite major differences in cellular composition:

(A) 100 cells each expressing 10 units of YAP (total = 1,000)

(B) 10 cells expressing 100 units of YAP and 90 cells expressing none (also total = 1,000)

While both tumors would exhibit the same overall YAP expression level in bulk RNA-seq data, their internal cellular composition is markedly different. We hypothesize that tumor (B), which exhibits greater heterogeneity, may be associated with a worse prognosis. Therefore,

we believe that bulk data cannot meaningfully address the question at hand, and that single-cell resolution is necessary to evaluate our hypothesis accurately. We have now discussed this point in the revised manuscript (**lines 328 - 339**).

3. The ferroptosis inducer data presented as part of rebuttal to point #5 should be done for at least PC-9 (that cell is being suggested based on the basal western blot data). It may be worthwhile, although optional, to do YAP/TAZ knockout any of those additional cells, but the basal validation of differential response to the ferroptosis inducers should be done and typically can be delivered within a very short time frame. If YAP is indeed different in the cells, this experiment should reflect differences in the way the cells respond to the inducers.

(Response) We appreciate the reviewer's suggestion and recognize the value of including PC-9 cells in the ferroptosis assay. We agree that investigating whether varying levels of YAP activity modulate ferroptosis sensitivity is an important question. However, we believe that a direct comparison between PC-9 cells (which exhibit higher phospho-YAP and therefore lower YAP activity) and LK-2 cells (which exhibit lower phospho-YAP and higher YAP activity) may not provide conclusive insights. Ferroptosis sensitivity is governed by multiple interconnected factors, particularly iron metabolism (e.g., levels of ferrous iron), lipid metabolism (e.g., PUFA/MUFA balance), and redox homeostasis (DOI: 10.1038/s41580-024-00703-5). Our current study specifically focuses on how YAP regulates the redox environment through BH4 production. If PC-9 and LK-2 cells shared similar iron and lipid metabolic profiles, any differences in ferroptosis sensitivity could more directly reflect differences in YAP activity. However, as these are distinct cell lines with likely intrinsic metabolic differences, attributing changes in ferroptosis sensitivity solely to YAP activity would be confounded.

A more suitable approach, in our view, is to evaluate ferroptosis responses within the same cell line under varying endogenous YAP activity levels. Since YAP activity can be modulated by factors such as cell density or serum conditions (DOI: 10.1101/gad.274027.115), this is experimentally feasible. Indeed, in a previous study (DOI: 10.1016/j.bbrc.2024.150373), we demonstrated that ferroptosis sensitivity correlates with endogenous YAP activity. Specifically, in Figure 2 of that work, we showed that sensitivity to cold-stress-induced ferroptosis is associated with YAP activation levels. Therefore, we believe that our current findings, in combination with our previously published data, provide sufficient evidence to support the conclusion that YAP/TAZ modulate ferroptosis sensitivity. We have now discussed this point in the revised manuscript (**lines 320 - 322**).

Response to Referee #2:

I appreciate the authors' efforts in performing additional experiments and providing further clarifications and discussions, which have addressed my previous critiques. Therefore, I support the acceptance of this manuscript for publication in EMBO Reports.

(Response) We sincerely thank the reviewer for their time and constructive comments, which have greatly contributed to improving our manuscript.

Response to Referee #3:

The authors have sufficiently addressed my points. I have no additional comments.

(Response) We sincerely thank the reviewer for their time and constructive comments, which have greatly contributed to improving our manuscript.

Dr. Toshiro MOROISHI
Institute of Science Tokyo
1-5-45 Yushima
Bunkyo-ku, Tokyo 113-8510
Japan

Dear Toshiro,

I am very pleased to accept your manuscript for publication in the next available issue of EMBO reports. Thank you for your contribution to our journal.

Kind regards,

Martina
